# CONFORMAL INDUCTIVE GRAPH NEURAL NETWORKS

**Soroush H. Zargarbashi**
CISPA Helmholtz Center for Information Security
zargarbashi@cs.uni-koeln.de

**Aleksandar Bojchevski**
University of Cologne
bojchevski@cs.uni-koeln.de

## ABSTRACT

Conformal prediction (CP) transforms any model's output into prediction sets guaranteed to include (cover) the true label. CP requires exchangeability, a relaxation of the i.i.d. assumption, to obtain a valid distribution-free coverage guarantee. This makes it directly applicable to transductive node-classification. However, conventional CP cannot be applied in inductive settings due to the implicit shift in the (calibration) scores caused by message passing with the new nodes. We fix this issue for both cases of node and edge-exchangeable graphs, recovering the standard coverage guarantee without sacrificing statistical efficiency. We further prove that the guarantee holds independently of the prediction time, e.g. upon arrival of a new node/edge or at any subsequent moment.

## 1 INTRODUCTION

Graph Neural Networks (GNNs) are used in many applications however without a reliable estimation of their output's uncertainty. We can not rely solely on the predicted distribution of labels $\pi(y \mid \boldsymbol{x})$ (e.g. from the softmax) as it is often uncalibrated. Therefore, it is crucial to find confidence estimates aligned with the true $p(y \mid \boldsymbol{x})$. There is a rich literature on uncertainty quantification methods many of which require retraining or modifications to the model architecture. Among them, almost none come with a guarantee, and many rely on the i.i.d. assumption. Given the interdependency structure (adjacency) we cannot easily adopt these methods for node classification. As a result, uncertainty quantification methods for GNNs are very limited (Stadler et al., 2021).

Conformal prediction (CP) is an alternative approach that uses the model as a black box and returns prediction *sets* guaranteed to cover the true label without any assumptions on the model's architecture or the data generating process. This guarantee is probabilistic and works for any user-specified probability $1 - \alpha$. To apply CP, we need a conformity score function $s : \mathcal{X} \times \mathcal{Y} \mapsto \mathbb{R}$ that quantifies the agreement between the input and the candidate label. Additionally, we need a held-out *calibration* set. The only assumption for a valid coverage guarantee is exchangeability between the calibration set and the test set. This makes CP applicable to non-i.i.d settings like transductive node classification – Zargarbashi et al. (2023) and Huang et al. (2023) showed that with a permutation-equivariant model (like almost all GNNs) CP obtains a valid coverage guarantee.

Given the dynamic nature of real-world graphs which evolve over time, *inductive* node classification is more reflective of actual scenarios than transductive. In the inductive setting, we are given an initial graph (used for training and calibration) which is progressively expanded by set of nodes and edges introduced to the graph over time. Importantly test nodes are not present at the training and calibration stage. For GNNs, updates in the graph cause an implicit shift in the embeddings of existing nodes and consequently the softmax outputs and conformity scores. Therefore, in the inductive setting, even assuming node/edge exchangeability (see § B for a formal definition), the coverage guarantee is no longer valid. Intuitively, calibration scores are no longer exchangeable with test scores, as soon as new nodes or edges are introduced (see Fig. 1 right).

To address this issue, Clarkson (2023) adapt the *beyond exchangeability* approach applying weighted CP for inductive node-classification aiming to recover the guarantee up to a bounded (but unknown) error (Barber et al., 2023). Unfortunately, this approach has an extremely limited applicability when applied with a realistic (sparse) calibration set. In sparse networks and limited labels, the method fails to predict *any* prediction set for a large number of nodes; and for the rest, the statistical efficiency is significantly low (see § C for details).

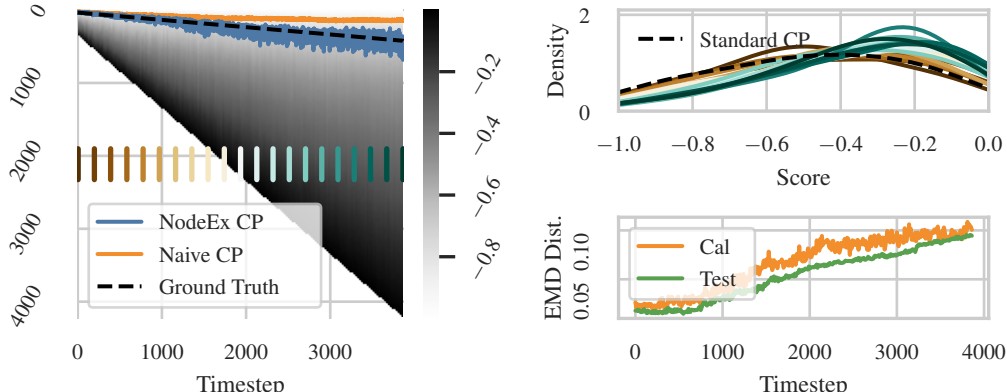

Figure 1: [Left] Each vertical line on the heatmap shows sorted true test scores at each timestep. The dashed line shows the true (unknown) $\alpha$-quantile and the quantile from each approach is also shown alongside. NodeEx CP (ours) closely tracks the true quantile, while naive CP deviates over time. [Upper right] Distributions from selected timesteps marked by the same color on the heatmap. The distribution shift is observable over time with new nodes appearing. [Lower right] The earth mover distance (EMD) between naive CP calibration scores and shifted true scores, denoted as "Test"; and EMD between naive and NodeEx CP scores, denoted as "Cal". Details in § D.5.

We show that for a node or edge exchangeable graph sequence (without distribution shift), we can adapt CP w.r.t. the implicit shift in score space, recovering the desired coverage guarantee. Node-exchangeability assumes that the joint distribution of nodes is invariant to any permutation. In the inductive setting, this means that any node is equally likely to appear at any timestep. We show that under node-exchangeability, the effect of new coming nodes is symmetric to the calibration set and the existing nodes. Hence, computing the conformity scores conditional to the subgraph at any timestep recovers the guarantee. Previous works under the transductive setting (Zargarbashi et al., 2023) and (Huang et al., 2023) also assume node-exchangeability.

While real-world graphs are sparse, all node-exchangeable generative processes produce either empty or dense graphs (see § 3). Therefore, we extend our study to the edge-exchangeable setting which can generate sparse graphs. In § 4.2 we show that w.r.t. scores, any edge-exchangeable sequence is equivalent to a weighted node-exchangeable sequence. Therefore, through weighted CP (weighted quantile lemma from Tibshirani et al. (2019)) we recover the $1 - \alpha$ coverage. Importantly, the improvement of our method is orthogonal to whether there is a distribution shift over time. Therefore, to address shift, we can still apply the weighting scheme from Barber et al. (2023) on top.

We focus on inductive node-classification. We define node-exchangeable (NodeEx) CP, and we show that (i) for node-exchangeable sequences, NodeEx CP obtains $1 - \alpha$ guarantee by calibrating over shifted conformal scores conditional to subgraphs in each timestep (Fig. 1 left), (ii) the $1 - \alpha$ guarantee is achievable via weighted CP on edge-exchangeable sequences, and (iii) The guarantee holds independent of the prediction time – the prediction set for a node at any timestep after its appearance benefits from the same coverage guarantee (e.g. prediction upon arrival or all nodes at once both result in the same coverage). We justify our approach with both theoretical and empirical results.

## 2 BACKGROUND: CONFORMAL PREDICTION AND TRANSDUCTIVE GNNS

Here we recall the general guarantee of CP (e.g. for image dataset). Through this paper, we assume a continuous score function without ties. This can apply to any function via adding noise. With weights $w_i \in [0, 1]$, and the sorting permutation $\tau^\star$, we define the weighted quantile Quant $(\cdot; \cdot; \cdot)$ as

$$\text{Quant}\left(\alpha; \{s_i\}_{i=1}^N; \{w_i\}_{i=1}^N\right) = \inf\left\{s_{\tau^\star(i)} : \frac{\sum_{j=1}^i w_{\tau^\star(j)}}{\sum_{j=1}^N w_j + 1} \geq \alpha\right\} \tag{1}$$

**Theorem 1** (Vovk et al. (2005)). *Let $\{(\boldsymbol{x}_i, y_i)\}_{i=1}^n$, and $(\boldsymbol{x}_{n+1}, y_{n+1})$ be exchangeable. With any continuous function $s : \mathcal{X} \times \mathcal{Y} \mapsto \mathbb{R}$ measuring the agreement between $\boldsymbol{x}$, and $y$, and user-specified significance level $\alpha \in (0, 1)$, with prediction sets defined as $\mathcal{C}_\alpha(\boldsymbol{x}_{n+1}) = \{y : s(\boldsymbol{x}_{n+1}, y) \geq \hat{q}\}$ where $\hat{q} := \mathrm{Quant}\,(\tilde{\alpha}; \{s(x_i, y_i)\}_{i=1}^n; 1)$. We have $\mathrm{Prob}\,[y_{n+1} \in \mathcal{C}(\boldsymbol{x}_{n+1})] \geq 1 - \alpha$.*

The marginal coverage probability is upper-bounded by $1 - \alpha + 1/(n + 1)$. With infinite samples, the coverage is distributed as a $\mathrm{Beta}(n + 1 - l, l)$ with $l = \lfloor (n + 1)\alpha \rfloor$ (Vovk, 2012), while with a fixed population $|\mathcal{D}| = n + m$, and an exchangeable calibration set $|\mathcal{D}_{\mathrm{cal}}| = n$, the probability of coverage $\mathrm{Cov}(\mathcal{D} \setminus \mathcal{D}_{\mathrm{cal}}) := (1/m)\mathbf{1}[y_i \in \mathcal{C}(\boldsymbol{x}_i)]_{i \in \mathcal{D} - \mathcal{D}_{\mathrm{cal}}}$ derives from a collection of hypergeometric distributions (Huang et al., 2023).

**Conformity scores**. The guarantee from Theorem 1 holds regardless of how the conformity score function $s : \mathcal{X} \times \mathcal{Y} \to R$ is defined. A wise choice of $s(\cdot, \cdot)$ is reflected in other metrics like the prediction set size. We use *adaptive* prediction sets (APS) with a score function defined as $s(\boldsymbol{x}, y) := -(\rho(\boldsymbol{x}, y) + u \cdot \pi(\boldsymbol{x})_y)$. Here $\rho(\boldsymbol{x}, y) := \sum_{c=1}^K \pi(\boldsymbol{x})_c \mathbf{1}\,[\pi(\boldsymbol{x})_c > \pi(\boldsymbol{x})_y]$ is the sum of all classes predicted as more likely than $y$, and $u \in [0, 1]$ is a uniform random value that breaks the ties between different scores to allow exact $1 - \alpha$ coverage (Stutz et al., 2022). Our method is independent of the choice of score function and we show its applicability on other (network-agnostic and network-aware) scores in § D.3.

**CP beyond exchangeability**. For cases where the calibration and test points are not exchangeable, there is a coverage gap $\Delta_\alpha$, i.e. $\mathrm{Prob}\,[y_{n+1} \in \mathcal{C}_\alpha(\boldsymbol{x}_{n+1})] \geq 1 - \alpha - \Delta_\alpha$. Barber et al. (2023) shows that $\Delta_\alpha$ has an upper bound that depends on how far the data deviates from exchangeability. This upper bound can be further controlled via weighted conformal prediction. For related works see § F.

## 2.1 CONFORMAL PREDICTION FOR TRANSDUCTIVE NODE-CLASSIFICATION

Consider a graph $\mathcal{G}(\mathcal{V}, \mathcal{E}, \boldsymbol{A}, \boldsymbol{y})$ with $\boldsymbol{X} \in \mathbb{R}^{n \times d}$ as node-features matrix, $\boldsymbol{A} \in \mathbb{R}^{n \times n}$ as adjacency matrix, and $\boldsymbol{y} \in \mathbb{R}^n$ as labels. $\mathcal{V} = \mathcal{V}_{\mathrm{tr}} \cup \mathcal{V}_{\mathrm{cal}} \cup \mathcal{V}_u$ is the set of vertices (training, calibration, and test). Let $f$ be a black-box permutation-equivariant model (e.g. a GNN) trained on labels of $\mathcal{V}_{\mathrm{tr}}$, and $s$ be a continuous score function with access to the outputs of $f$. The score function $s$ may or may not use information about the graph structure (see § D.3). In the transductive setup the graph $\mathcal{G}$ is fixed, and training and calibration is performed with all observed nodes (including features and structure). The labels of $\mathcal{V}_{\mathrm{tr}}$ and $\mathcal{V}_{\mathrm{cal}}$ are used respectively for training and calibration, while all other labels remain unseen. Here when $\mathcal{V}_{\mathrm{cal}}$ and $\mathcal{V}_u$ are exchangeable, standard CP can be applied (Zargarbashi et al., 2023; Huang et al., 2023). For a set with a fixed number of nodes $\mathcal{V}'$, we define the discrete coverage as $\mathrm{Cov}(\mathcal{V}') = \frac{1}{|\mathcal{V}'|} \sum_{v_j \in \mathcal{V}'} \mathbf{1}\left[y_j \in \hat{\mathcal{C}}(v_j)\right]$. The following theorem shows that under transductive calibration, CP yields valid prediction sets.

**Theorem 2** (Rephrasing of Theorem 3 by Huang et al. (2023)). *For fixed graph $\mathcal{G}$, and a permutation-equivariant score function $s(\cdot, \cdot)$, with $\mathcal{V}_{\mathrm{cal}} = \{(v_i, y_i)\}_{i=1}^N$ exchangeably sampled from $\mathcal{V} \setminus \mathcal{V}_{\mathrm{tr}}$, and prediction sets $\hat{\mathcal{C}}(v) = \left\{y : s(v, y) \geq \mathrm{Quant}\left(\alpha; \{s(v_i, y_i)\}_{i=1}^N ; 1\right)\right\}$ we have*

$$\mathrm{Prob}\,[\mathrm{Cov}(\mathcal{V}_u) \leq t] = 1 - \Phi_{HG}(i_\alpha - 1; M + N, N, \lceil Mt \rceil + i_\alpha) \tag{2}$$

*where $i_\alpha = \lceil (N + 1)(1 - \alpha) \rceil$ is the unweighted $\alpha$-quantile index of the calibration scores, and $\Phi_{HG}(\cdot; N, n, K)$ is the c.d.f. of a hyper-geometric distribution with parameters $N$ (population), $n$ number of samples, and $K$ number of successful samples among the population.*

**The issue with inductive node-classification**. The above approach does not directly translate to the inductive setting. As soon as the graph is updated, the implicit shift in the nodes embeddings results in a distribution shift w.r.t. the calibration scores that are computed before the update. This breaks the exchangeability as shown by Zargarbashi et al. (2023) and in Fig. 1. To address this issue, Clarkson (2023) adopts weighted CP (Barber et al., 2023) with neighborhood-dependent weights (NAPS), limiting the calibration nodes to those inside an immediate neighborhood of a test node. Applying NAPS on sparse graphs or with small (realistic) calibration sets, leaves a significant proportion of test nodes with an "empty" calibration set. Hence, its applicability is limited to very special cases (see § C for details). Moreover, NAPS does not quantify the coverage gap $\Delta_\alpha$. In contrast, when assuming either node or edge-exchangeability, the gap for our approach is zero regardless of the weights.

## 3  INDUCTIVE GNNS UNDER NODE AND EDGE EXCHANGEABILITY

In the inductive scenario, the graph changes after training and calibration. Therefore, the model $f$ is trained, and CP is calibrated only on a subgraph before the changes. We track these updates in a sequence of graphs $\mathcal{G}_1, \mathcal{G}_2, \dots$ where $\mathcal{G}_t = (\mathcal{V}_t, \mathcal{E}_t)$ is a graph with a finite set of vertices and edges. We focus on progressive updates meaning $\forall t : \mathcal{V}_t \subseteq \mathcal{V}_{t+1}$ and $\mathcal{E}_t \subseteq \mathcal{E}_{t+1}$. At timestep $t$ in a node-inductive sequence, the update adds a node $v_t$ with all its connections to existing vertices. In an edge-inductive sequence, updates add an edge, which may or may not bring unseen nodes with it. W.l.o.g update sets are singular. Node-inductive and edge-inductive sequences are formally defined in § B.

We call a node-inductive sequence to be *node-exchangeable* if the generative process is invariant to the order of nodes, meaning that any permutation of nodes has the same probability. Analogously, a sequence is *edge-exchangeable* if all permutations of edges are equiprobable. For both cases, the problem is node-classification. Since split CP is independent of model training w.l.o.g. we assume that we train on an initial subgraph $\mathcal{G}_0$. We do not need to assume that $\mathcal{G}_0$ is sampled exchangeably. Formal definitions of node-exchangeability and edge-exchangeability are provided in § B.

For transductive GNNs, both Zargarbashi et al. (2023) and Huang et al. (2023) assume that the calibration set is sampled node-exchangeably w.r.t. the test set, however the entire graph is fixed and given. In contract, we assume that the sequence of graphs $\mathcal{G}_1, \mathcal{G}_2, \dots$ itself, including calibration and test nodes, is either node- or edge-exchangeable. Transductive setting can be seen as a special case.

**Sparsity**. Any node-exchangeable graph is equivalent to mixture of *graphons* (Aldous, 1981; Hoover, 1979; Cai et al., 2016). A graphon (or graph limit) is a symmetric measurable function $W : [0,1]^2 \mapsto [0,1]$ and the graph is sampled by drawing $u_i \sim \text{Uniform}[0,1]$ for each vertex $v_i$, and an edge between $v_i$, and $v_j$ with probability $W(u_i, u_j)$. We know that graphs sampled from a graphon are almost surely either empty or dense (Orbanz & Roy, 2014; Cai et al., 2016) – the number of edges grows quandratically w.r.t. the number of vertices. However, sparse graphs (with edges sub-quadratical in the number of vertices) are more representative to real-world networks. Therefore, we also consider edge-exchangeable graph sequences (Cai et al., 2016) that can achieve sparsity.

## 4  CONFORMAL PREDICTION FOR EXCHANGEABLE GRAPH SEQUENCES

The first $N$ nodes of the sequence (and their labels) are taken as the calibration set, leaving the rest of the sequence to be evaluated.[1] For easier analysis, we record the coverage of each node at each timestep. Let $T_0$ be the time when the last calibration node arrived. Up to timestep $T$, let matrix $C \in \{0,1\}^{\mathcal{V}_T \times T}$ with $C[i,t]$ indicate whether the prediction set for test node $v_i$ at timestep $t$ covers the true label $y_i$. The time index in $C$ is relative to $T_0$ and w.l.o.g. we index nodes upon their appearance. Hence, in each column $C[\cdot, t]$, the first $|\mathcal{V}_t|$ elements are in $\{0,1\}$ and the rest are N/A. Formally:

$$C[i,t] = \begin{cases} \mathbf{1}\left[y_i \in \left\{y : s(v_i, y) \geq \text{Quant}\left(\alpha; \{s(v_i, y_i | \mathcal{G}_t)\}_{v_i \in \mathcal{V}_{\text{cal}}}; 1\right)\right\}\right] & i \geq t \\ \text{N/A} & \text{o.w.} \end{cases} \tag{3}$$

Here we assume that one can evaluate a node at any timestep after its appearance. We define $\mathcal{V}_{\text{eval}}^{(t)} \subseteq \mathcal{V}_t$ as the set of nodes evaluated at timestep $t$. For an edge-exchangeable sequence, a node $v$ is considered as active (existing) upon the arrival of the first edge connected to it. After $T$ timesteps, we call the (recorded) sub-partition $\cup_{i=1}^{T} \mathcal{V}_{\text{eval}}^{(i)}$ as an evaluation mask $\mathbb{I}_T$. Any node appears at most once in $\mathbb{I}_T$ (we do not re-evaluate a node). We define the $\text{Cov}(\mathcal{V}_{\text{eval}}^{(t)}) = (1/|\mathcal{V}_{\text{eval}}^{(t)}|) \sum_{v_i \in \mathcal{V}_{\text{eval}}^{(t)}} C[i,t]$ as the empirical coverage over the mask $\mathcal{V}_{\text{eval}}^{(t)}$. Similarly, $\text{Cov}(\mathbb{I}_T)$ is the empirical coverage for $\mathbb{I}_T$ (average over all records). We visualize $1 - C$ under node exchangeability on Fig. 2.

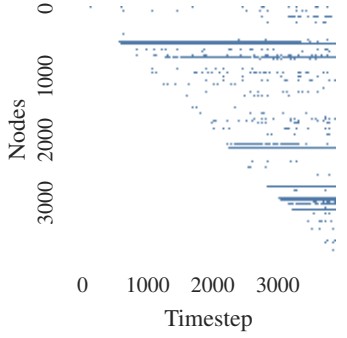

Figure 2: $1 - C$ for Cora. Details in § B

---

[1] All results can be trivially generalized to the case where the calibration nodes are scattered in the sequence. This only affects the width of the coverage interval (the variance) as it is directly a function of the number of available calibration nodes when computing prediction sets.

## 4.1 Node-exchangeable Sequences

A node-inductive graph sequence is node-exchangeable if any permutation of node appearance is equally likely – at each step, any unseen node has the same probability of appearing (see § B).

With the arrival of new nodes, the distribution of embeddings (and consequently scores) shifts (as shown in Fig. 1). This is why calibrating prior to the update fails to guarantee coverage for new nodes. However, with node-exchangeability, calibration and evaluation scores computed conditional to a specific subgraph are still exchangeable. We extend Theorem 2 to inductive GNNs on node-exchangeable graph sequences for any subset of evaluated vertices.

**Proposition 1.** *At time step $t$, with graph $\mathcal{G}_t = (\mathcal{V}_t, \mathcal{E}_t)$ from a node-exchangeable sequence, and a calibration set $\mathcal{V}_{\mathrm{cal}}$ consisting of the first $n$ nodes in $\mathcal{G}_t$, for any $v_j \in \mathcal{V}_{\mathrm{eval}}^{(t)} \subseteq \mathcal{V}_t \setminus \mathcal{V}_{\mathrm{cal}}$ define*

$$\mathcal{C}^{(t)}(v_j) = \left\{ y : s(v_j, y \mid \mathcal{G}_t) \geq \mathrm{Quant}\left( \alpha; \{ s(v_i, y_i \mid \mathcal{G}_t) \}_{i \in \mathcal{V}_{\mathrm{cal}}}; 1 \right) \right\} \tag{4}$$

*Then* $\mathrm{Prob}\left[ y_j \in \mathcal{C}^{(t)}(v_j) \mid v_j \in \mathcal{V}_{\mathrm{eval}}^{(t)} \right] \geq 1 - \alpha$. *Moreover with $m := |\mathcal{V}_{\mathrm{eval}}^{(t)}|$,*

$$\mathrm{Prob}\left[ \mathrm{Cov}(\mathcal{V}_{\mathrm{eval}}^{(t)}) \leq \beta \right] = 1 - \Phi_{\mathrm{HG}}(i_\alpha - 1; m + n, n, i_\alpha + \lceil \beta t \rceil) \tag{5}$$

We defer proofs to § B. Here we provide an intuitive justification of the theorem. First, if $\mathcal{V}_{\mathrm{eval}}^{(t)}$ includes all vertices the problem reduces to the transductive case, for which Theorem 2 applies with scores conditional on the current graph. Otherwise, $\mathcal{V}' := \mathcal{V}_t \setminus \mathcal{V}_{\mathrm{eval}}^{(t)}$ is not empty. Here the effect of $\mathcal{V}'$ is symmetric to $\mathcal{V}_{\mathrm{cal}}$ and $\mathcal{V}_{\mathrm{eval}}^{(t)}$. For instance, consider a linear message passing $\boldsymbol{Z} = \boldsymbol{AXW}$ with weight matrix $\boldsymbol{W}$ (our results hold in general). From standard CP we know that for row-exchangeable matrix $\boldsymbol{X}$, $\boldsymbol{XW}$ is also row-exchangeable (a linear layer is permutation equivariant). Hence, we just question the row-exchangeability of $\boldsymbol{AX}$. Split $\boldsymbol{X}$ into $[\boldsymbol{X}_{\mathrm{cal}}^\top \mid \boldsymbol{X}_{\mathrm{eval}}^\top \mid \boldsymbol{X}_{\mathcal{V}'}^\top]^\top$, and similarly split $\boldsymbol{A}$ into nine blocks based on the endpoints of each edge. We have that $\boldsymbol{AX}$ equals

$$\begin{bmatrix} \boldsymbol{A}_{\mathrm{cal}\cdot\mathrm{cal}} & \boldsymbol{A}_{\mathrm{cal}\cdot\mathrm{eval}} & \boldsymbol{A}_{\mathrm{cal}\cdot\mathcal{V}'} \\ \boldsymbol{A}_{\mathrm{eval}\cdot\mathrm{cal}} & \boldsymbol{A}_{\mathrm{eval}\cdot\mathrm{eval}} & \boldsymbol{A}_{\mathrm{eval}\cdot\mathcal{V}'} \\ \boldsymbol{A}_{\mathcal{V}'\cdot\mathrm{cal}} & \boldsymbol{A}_{\mathcal{V}'\cdot\mathrm{eval}} & \boldsymbol{A}_{\mathcal{V}'\cdot\mathcal{V}'} \end{bmatrix} \boldsymbol{X} = \underbrace{\begin{bmatrix} \boldsymbol{A}_{\mathrm{cal}\cdot\mathrm{cal}}\boldsymbol{X}_{\mathrm{cal}} + \boldsymbol{A}_{\mathrm{cal}\cdot\mathrm{eval}}\boldsymbol{X}_{\mathrm{eval}} \\ \boldsymbol{A}_{\mathrm{eval}\mathrm{cal}}\boldsymbol{X}_{\mathrm{cal}} + \boldsymbol{A}_{\mathrm{eval}\cdot\mathrm{eval}}\boldsymbol{X}_{\mathrm{eval}} \\ \cdots \end{bmatrix}}_{\mathrm{Mat}(1)} + \underbrace{\begin{bmatrix} \boldsymbol{A}_{\mathrm{cal}\cdot\mathcal{V}'}\boldsymbol{X}_{\mathcal{V}'} \\ \boldsymbol{A}_{\mathrm{eval}\cdot\mathcal{V}'}\boldsymbol{X}_{\mathcal{V}'} \\ \cdots \end{bmatrix}}_{\mathrm{Mat}(2)}$$

For the guarantee to be valid, the first $|\mathcal{V}_{\mathrm{cal}}| + |\mathcal{V}_{\mathrm{eval}}|$ rows should be exchangeable. For Mat(1), this already holds due to node-exchangeability and Theorem 2. With $\boldsymbol{X}_{\mathcal{V}'}$ being common in both blocks of Mat(2), we only need $\boldsymbol{A}_{\mathrm{cal}\cdot\mathcal{V}'}$ and $\boldsymbol{A}_{\mathrm{eval}\cdot\mathcal{V}'}$ to be row-exchangeable which again holds due to node exchangeability. Hence, the effect of $\mathcal{V}'$ is symmetric to calibration and evaluation sets. In other words, the shifted embedding still preserves the guarantee conditional to the graph at timestep $t$. The only requirement is to compute the conformal scores for all nodes including the calibration set and *dynamically update* the quantile threshold – the threshold depends on $t$. Another way to understand the theorem is that for any column in $\boldsymbol{C}$, the expectation of any subset of elements $\geq 1 - \alpha$. Next, we generalize this guarantee to any evaluation mask independent of time.

**Theorem 3.** *On a node-exchangeable graph sequence, with exchangeably sampled $\mathcal{V}_{\mathrm{cal}}$ consisting of the first $n$ nodes in $\mathcal{G}_t$, for any valid partition $\mathbb{I}_T = \cup_{i=1}^T \mathcal{V}_{\mathrm{eval}}^{(i)}$ we have*

$$\mathrm{Prob}\left[ y_i \in \mathcal{C}^{(t)}(v_i) \mid v_i \in \mathcal{V}_{\mathrm{eval}}^{(t)} \right] \geq 1 - \alpha \tag{6}$$

*Moreover, it holds that* $\mathrm{Prob}\left[ \mathrm{Cov}(\mathbb{I}_T) \leq \beta \right] = 1 - \Phi_{\mathrm{HG}}(i_\alpha; |\mathbb{I}_T| + n, n, i_\alpha + \lceil |\mathbb{I}_T| \cdot \beta \rceil)$

Theorem 3 indicates that CP applies to inductive GNNs conditional to the subgraph in which each node is evaluated. We will leverage this insight in § 5. In conclusion, with a node-exchangeable graph sequence, for any node $v_i$ appearing at timestep $t$, and evaluated at any timestep $t' \geq t$, it holds that $\mathrm{Prob}\left[ y_i \in \mathcal{C}^{(t')}(v_i) \right] \geq 1 - \alpha$. Two special cases of this result are evaluation upon appearance (the diagonal of $\boldsymbol{C}$) and evaluation all at once (the last column of $\boldsymbol{C}$) which follows from Theorem 2. The latter is also equivalent to the simultaneous-inductive setting in Zargarbashi et al. (2023). The user is flexible to delay the prediction to any time, still preserving the guarantee.

## 4.2 EDGE-EXCHANGEABLE SEQUENCES

In an edge-inductive sequence at each timestep an edge is added, $\mathcal{G}_{t+1} = (\mathcal{V}_t \cup V(e_{t+1}), \mathcal{E}_t \cup e_{t+1})$.[2] This edge may introduce new nodes or connect existing ones. When all permutations $\mu$ of the edge-sequence are equally likely, $\text{Prob}\left[(e_1, \ldots e_m)\right] = \text{Prob}\left[(\mu(e_1), \ldots \mu(e_m))\right]$, the sequence is edge-exchangeable (see § B for a formal definition). We address this setting using a special case of weighted quantile lemma (Tibshirani et al., 2019) with weights defined by the frequency of elements.

**Lemma 1.** *Let $\mathcal{X} = \{x_1, \cdots, x_m\}$ be exchangeable random variables and $f : 2^{\mathcal{X}} \mapsto \mathbb{R}$ be a mapping defined on subsets of $\mathcal{X}$. For any partitioning $\cup_{i=1}^{n+1} \mathcal{X}_i = \mathcal{X}$ and $z_i := f(\mathcal{X}_i)$ we have:*

$$\text{Prob}\left[z_{n+1} \leq \text{Quant}\left(\beta; \{z_i\}_{i=1}^{n} \cup \{\infty\}; \left\{\frac{1}{|\mathcal{X}_i|}\right\}_{i=1}^{n+1}\right)\right] \geq \beta$$

We introduce the edge-exchangeable (EdgeEx) CP and prove its guarantee by showing that at any timestep $t$, an edge exchangeable sequence, is decomposed into weighted node exchangeable subsequences with weights equal to $1/\deg(v)$. Then, on each subsequence weighted CP maintains a valid guarantee. In this setup, there are no isolated nodes, any node can be evaluated upon appearance.

**Theorem 4.** *At each timestep $t$, given graph $\mathcal{G}_t = (\mathcal{V}_t, \mathcal{E}_t)$ from an edge-exchangeable sequence and with a calibration set $\mathcal{V}_{\text{cal}}$, define $q = \text{Quant}\left(\alpha; \{s_i\}_{v_i \in \mathcal{V}_{\text{cal}}}; \left\{1/\deg(v_i)_{v_i \in \mathcal{V}_{\text{cal}}}\right\}\right)$, and for any $v_j \in \mathcal{V}_{\text{eval}}^{(t)}$ define $\mathcal{C}^{(t)}(v_j) := \{y : s(v_j, y) \geq q\}$. We have $\text{Prob}\left[y_j \in \mathcal{C}^{(t)}(v_j)\right] \geq 1 - \alpha$.*

Via Theorem 4, EdgeEx CP is guaranteed to provide $1 - \alpha$ coverage. The result holds for each timestep $t$, conditional to that timestep.

## 5 NODE EXCHANGEABLE AND EDGE EXCHANGEABLE CP

Building upon the theory in § 4, for node-exchangeable graph sequences we define node-exchangeable (NodeEx) CP with coverage guarantee conditional to the subgraph at each timestep. Recall that the shift in scores upon changes in the graph is symmetric for calibration and evaluation nodes. Hence, NodeEx recomputes the calibration scores with respect to the additional context. In an edge-exchangeable sequence via Theorem 4, EdgeEx CP – weighted NodeEx CP with $w_i = 1/\deg(v_i)$, results in a similar valid guarantee. Any further details about NodeEx also generalizes to the EdgeEx via Theorem 4.

With both settings at each timestep $t$ we have seen a set of nodes $\mathcal{V}_t$. Let $\mathcal{V}_{\text{eval}}^{(t)} \subset \mathcal{V}_t$ be the set of nodes evaluated at timestep $t$. This set can contain any node from the past as long as they are not already evaluated. Specifically, we cannot use the prediction sets of a particular node multiple times to compare and pick a specific one (see § D.1 for a discussion). We always assume that the calibration set is the union of all nodes appeared up to timestep $t = T_0$ and the test set contains the remaining nodes. For a sequence of timesteps and corresponding evaluation sets $\left\{(t_i, \mathcal{V}_{\text{eval}}^{t_i})\right\}_{i=T_0+1}^{T}$ where $\cup_{i=T_0+1}^{T} \mathcal{V}_{\text{eval}}^{t_i} = \mathcal{V}_T - \mathcal{V}_{\text{cal}}$, NodeEx CP (see § A for algorithm) returns valid prediction sets. We use EdgeEx CP for edge-exchangeable sequences.

**CP on node-conditional random subgraphs**. Another application of Proposition 1 is to produce subgraph conditional prediction sets even for transductive node-classification. For each node $v_{\text{test}}$, we define $K$ subgraphs each including $\{v_{\text{test}}\}$, $\mathcal{V}_{\text{cal}}$ and randomly sampled $\mathcal{V}' \subset \mathcal{V}_{\mathcal{G}}$. Then, we average the scores and return prediction sets. In another approach, we can run $K$ concurrent CPs, each resulting in a prediction set. Then we combine them with a *randomized* voting mechanism to a final prediction set. The resulting prediction sets include true labels with $1 - \alpha$ probability. This approach works for any number of subgraphs. See § D.1 for further explanations.

**Relation to full conformal prediction**. In our paper, we technically adopted the split conformal[3] approach where model training is separate from calibration due to its computational efficiency. An alternative is full conformal prediction, where to obtain a score for a (calibration) node $v_i \in \mathcal{V}_{\text{cal}}$ and any candidate label $y'$ w.r.t. a test node $v_{\text{test}}$ we train a model from scratch on $\mathcal{V}_{\text{cal}} \cup \{(v_{\text{test}}, y')\}$ and

---

[2]Similar results apply for updates with more than one edge.

[3]Sometimes called inductive conformal prediction. However, this is an orthogonal use of the word inductive.

consider its prediction for $v_i$. This involves training $|\mathcal{V}_{\text{cal}}| + 1$ models for each $y'$ and each test node, which is extremely expensive but can use all[4] available labels. Transductive node classification can be seen as a middle ground between split and full conformal as explained by Huang et al. (2023). Our approach is similarly in between – instead of retraining the model from scratch we simply update the embeddings (and thus scores) by performing message-passing with the newly arrived nodes/edges.

# 6 EXPERIMENTAL EVALUATION

## 6.1 EXPERIMENTAL SETUP AND DISCUSSION OF METRICS

We evaluate our approach for both node and edge exchangeable sequences where we have three options to make predictions: (i) upon node arrival, (ii) at a given fixed timestep shared for all nodes, (iii) at an arbitrary node-specific timestep (we choose at random). We compare our approach with naive CP where the coverage guarantee will not hold. Later we discuss why both over- and under-coverage are invalid. For completeness, we compare NodeEx CP with NAPS in § C in addition to a detailed overview of its drawbacks. However, as discussed in § 2.1, NAPS is not a suitable baseline. We use APS (Romano et al., 2020) as the base score function while other scores are tested in § D.

**Models and datasets.** We consider 9 different datasets and 4 models: GCN Kipf & Welling (2017), GAT Veličković et al. (2018), and APPNP Klicpera et al. (2019) as structure-aware and MLP as a structure-independent model. We evaluate our NodeEx, and EdgeEx CP on the common citation graphs CoraML McCallum et al. (2004), CiteSeer Sen et al. (2008), PubMed Namata et al. (2012), Coauthor Physics and Coauthor CS Shchur et al. (2018), and co-purchase graphs Amazon Photos and Computers McAuley et al. (2015); Shchur et al. (2018) (details in § E). Results for Flickr (Young et al., 2014), and Reddit2 (Zeng et al., 2019) (with GCN model and their original split) datasets are in § D.5. Here, the model only affects the efficiency and not the validity.

**Evaluation procedure**. For any of the mentioned datasets, we sample 20 nodes per class for training and 20 nodes for validation with stratified sampling. For the node-exchangeable setup, the calibration set has the same size as the training. For the edge-exchangeable sequence, the same number of *edges* are sampled. Therefore, each round of simulation has a potentially different number of calibration nodes for the edge-exchangeable setup. First, we take a random sample of nodes as train/val set and train the model on the resulting subgraph $\mathcal{G}_0$. Then, for the node-exchangeable sequence, we first sample a calibration set randomly from the remaining nodes. Then, at each timestep, we add a random unseen node to the graph (with all edges to the existing nodes) and predict the class probability given the updated subgraph. We use the updated conformal scores to create prediction sets for *all* existing test nodes until time $t$ and record the empirical coverage results in column $t$ of the coverage matrix $\boldsymbol{C}$. Similarly, for the edge-exchangeable sampling, we sample calibration *edges*, and take both ends as calibration nodes. The remaining edges are sampled one at a time. The rest of the procedure is the same as node-exchangeable setting and results in an analogous $\boldsymbol{C}$.

**Challenges with small calibration sets**. With a limited number of samples in the calibration set, there will be an additional error in the coverage due to the discrete quantile function with a low sample rate. In practice, we take the $\lfloor n/(n+1) \cdot \alpha \rfloor$ index of a discrete array as the conformal threshold which is often not exactly equal to $1 - \alpha$ quantile in the continuous domain. Therefore, we expect $1/(|\mathcal{V}_{\text{cal}}| + 1)$ error around $1 - \alpha$. This error is in addition to the variance of the Beta or Hyper-geometric distribution and will converge to 0 with increasing calibration set size.

**Distance from target coverage**. Our main evaluation criteria is the distance of empirical coverage w.r.t. the target $1 - \alpha$. For all datasets we set the desired coverage to $90\%$, and we report the (absolute) distance w.r.t. this value (see § E). Each reported result is an average of 10 different random node-exchangeable sequences and 15 different random edge-exchangeable sequences. Note that due to homophily, we observe a higher empirical coverage in the non-exchangeable case which can wrongly be interpreted as being better. To address any potential confusion, first the guarantee is invalid if it breaks either the lower or the upper-bound. In real-world deployment we do not have ground-truth labels to calculate empirical coverage, and if the guarantee is broken (in either direction) the output is unreliable – nullifying the main goal of CP. Second, a higher empirical coverage also results in larger set sizes making the prediction sets less useful (see also Fig. 4).

---

[4]This is unlike split conformal that needs to split the available labels into training and calibration sets.

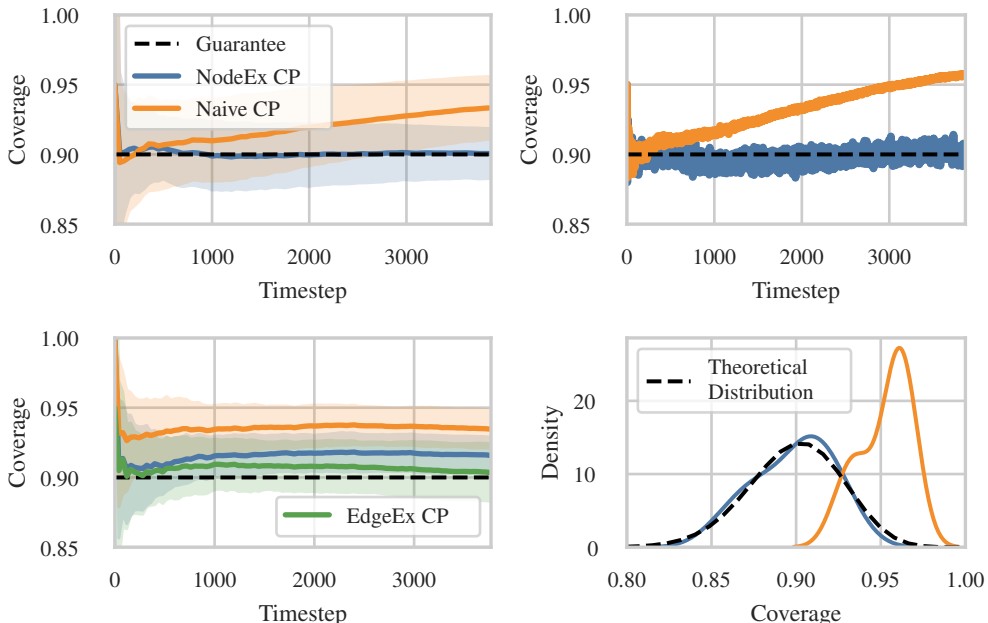

Figure 3: [Upper left] Coverage over time under node exchangeability when predicting upon node arrival (diagonals of $C$). [Upper right] Coverage when we instead predict at a fixed time (columns of $C$). [Lower right] Same as upper right but under edge-exchangeability. [Lower left] The empirical distribution of coverage when we predict at node-specific times (fixed entries of $C$), compared to the theoretical distribution. Sample size results in a slight *expected* shift. The transparent lines show a particular sequence, and the thick solid lines shows the average over 10 (15) sequences.

**Efficiency and singletons**. In addition to coverage, which is the main goal of our work, we also briefly study the efficiency (average set size) and the singleton hit – fraction of correct sets of size 1. As we will see in § 6.2, our NodeEx and EdgeEx CP improve these metrics as a byproduct. Some scoring functions such as RAPS (Angelopoulos et al., 2021) and DAPS (Zargarbashi et al., 2023) directly target these metrics and can be used on top of our approach. We explore DAPS in § D.3.

## 6.2 Evaluation of Empirical Coverage, Set Size and Singleton Hit

Table 1 shows the deviation from the coverage guarantee for different datasets when the label of each node is predicted *upon its arrival*. As shown in the table, while naive CP shows a significant shift from the $1 - \alpha = 0.9$ coverage, NodeEx CP maintains the empirical coverage close to the desired value. In Fig. 3 (upper left) we show the temporal evolution of coverage. We show the coverage for all nodes that arrived until time $t$ and see that the guarantee is preserved for each timestep $t$. As mentioned before, the goal is to achieve near $1 - \alpha$ empirical coverage, the over-coverage of standard CP might misleadingly appear as a better result. Not only does over-coverage come at the cost of less efficiency as shown in Fig. 4 (see § D.2 for details), but the empirical coverage of non-guaranteed prediction sets is not predictable apriori.

Fig. 3(lower left) shows the same experiment for edge-exchangeability. While NodeEx CP guarantees coverage under node-exchangeability, the weights specified in our EdgeEx CP are necessary for the guarantee to hold on an edge-exchangeable sequence. Standard CP fails again.

As explained in § 4 any node can be evaluated at *any timestep* after its appearance. Fig. 3 (upper right) shows the result when predicting at a fixed time (instead of upon arrival). Additionally, Fig. 3 (lower right) shows the empirical distribution of coverage for node-specific times (random subsets of nodes, each node predicted at a random time after appearance). For NodeEx CP, the empirical distribution matches the theoretical one, while the coverage of naive CP substantially diverges.

| Dataset | Acc | Node Exch. Sequence | | Edge Exch. Sequence | | |
| --- | --- | --- | --- | --- | --- | --- |
| | | NodeEx CP | Naive CP | EdgeEx CP | NodeEx CP | Naive CP |
| Cora-ML | 0.800 | **0.280** | 5.860 | **1.929** | 3.883 | 6.860 |
| PubMed | 0.777 | **1.254** | 4.649 | **1.241** | 3.405 | 5.315 |
| CiteSeer | 0.816 | **0.039** | 4.150 | **0.335** | 1.572 | 3.460 |
| Coauth-CS | 0.914 | **0.397** | 4.082 | **3.024** | 4.662 | 7.835 |
| Coauth-Phy. | 0.940 | **0.555** | 2.689 | **2.240** | 4.378 | 6.123 |
| Amz-Comp. | 0.788 | **0.263** | 6.373 | **2.687** | 5.727 | 7.036 |
| Amz-Photo | 0.868 | **0.127** | 3.483 | **2.546** | 4.130 | 6.613 |

Table 1: Average absolute deviation from guarantee (in %, for `GCN` model and 1 train/val split).

Unsurprisingly, under node exchangeability, in Fig. 4 we see that our NodeEx CP improves both additional metrics – has smaller sets and larger singleton hit ratio. The same holds for other settings.

In all experiments, we consider a sparse (and thus realistic) calibration set size, e.g. for `PubMed` we sample 60 nodes for calibration, which is a significantly lower number compared to other tasks like image classification with 1000 datapoints for same purpose (Angelopoulos et al., 2021). The calibration size controls the concentration (variance) around $1 - \alpha$ as reflected by the transparent lines on Fig. 3. Increasing the set size reduces the variance but the mean is always $1 - \alpha$.

**Limitations**. We identified three main limitations. First, the guarantee is marginal. Second, real-world graphs may not satisfy node- or edge-exchagability. This can be partially mitigated by the beyond exchangeability framework. Finally, the guarantee does not hold for adversarially chosen evaluation sets $\mathcal{V}_{\text{eval}}$. In other words, the choice of which nodes is evaluated at which timesteps must be prior to observing the prediction set. We provide a longer discussion on each of these limitations in § A. Our main focus is on the validity. However, we also reported other metrics such as the average set size, singleton hit ratio and other score functions in § D.

## 7 CONCLUSION

We adapt conformal prediction to inductive node-classification for both node and edge exchangeable graph sequences. We show that although introducing new nodes/edges causes a distribution shift in the conformity scores, this shift is symmetric. By recomputing the scores conditional to the evaluation subgraph, we recover the coverage guarantee. Under edge-exchangeability, we need to also account for the node degrees to maintain validity. Importantly, our approach affords flexibility – the guarantee holds regardless of prediction time which can be chosen differently for each node.

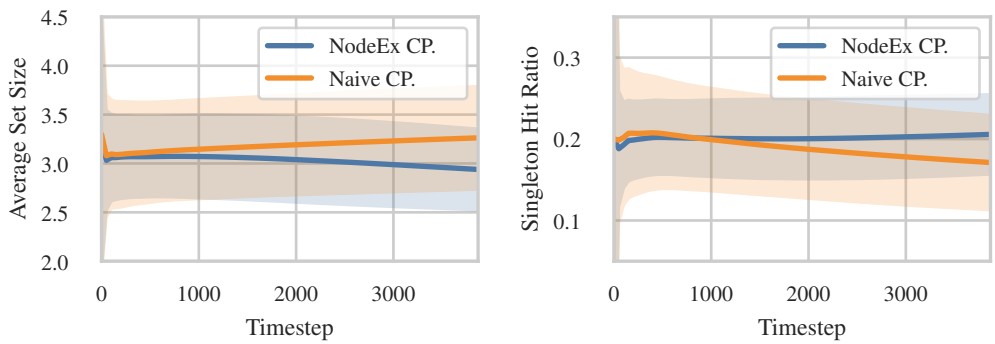

Figure 4: [Left] Average set size (lower is better) and [Right] singleton hits ratio (higher is better) of naive CP vs. NodeEx CP for `CiteSeer` and `GCN`. Our approach improves both metrics.

## REPRODUCIBILITY STATEMENT

All datasets of our experiments are publicly available and referenced in § 6.1 in addition to details on experimental setups. The pseudocode of our approach is provided in § A. The hyperparameters of models are given in § E. The code of experiments is also uploaded as supplementary material.

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

## A ALGORITHM FOR NODEEX CP AND WEIGHTED EDGEEX CP

At each timestep $t$ with node-exchangeable graph $\mathcal{G}_t$, we first run the GNN model to extract conformity scores for all nodes in $\mathcal{V}_t$. We compute the $\alpha$-quantile of the conformity scores conditional to the current graph, and for each new test node $v_{\text{test}} \in \mathcal{V}_{\text{eval}}^{(t)}$ we return any candidate label with a score larger than the conditional threshold. For an edge-exchangeable graph, the algorithm is still the same unless we compute the conditional weighted quantile with weights equal to $1/d(v)$ for all calibration vertices $v$. Algorithm 1 shows the pseudocode for NodeEx CP. The code is accessible at `github.com/soroushzargar/conformal-node-classification`.

---

**Algorithm 1:** NodeEx and EdgeEx CP for inductive node classification

---

**Data:** Graph $\mathcal{G}_t$ at timestep $t$.
Calibration vertex set $\mathcal{V}_{\text{cal}}$,
Permutation equivariant score function $s$ (e.g. a function of GNN),
Evaluation vertex set $\mathcal{V}_{\text{eval}}^{(t)}$
**Result:** $C^t(v_j) : v_j \in \mathcal{V}_{\text{eval}}^{(t)}$
1   $\boldsymbol{S}_t \leftarrow s(\mathcal{G}_t)$ ;         `// Get score functions for all vertices`
2   **if** *node-exchangeable sequence* **then**
3      Set $\tau_t \leftarrow \text{Quant}\left(\alpha; \{\boldsymbol{S}_t[i, y_i]\}_{v_i \in \mathcal{V}_{\text{cal}}}; 1\right)$;
4   **else if** *edge-exchangeable sequence* **then**
5      Set $\tau_t \leftarrow \text{Quant}\left(\alpha; \{\boldsymbol{S}_t[i, y_i]\}_{v_i \in \mathcal{V}_{\text{cal}}}; 1/\text{deg}(v_i \mid \mathcal{G}_t)\right)$;
6   $\forall v_j \in \mathcal{V}_{\text{eval}}^{(t)}$, set $C^t(v_j) = \{y : \boldsymbol{S}_t[j, y] \geq \tau^t\}$ ;

---

**Naive CP**. In naive CP we compute calibration scores before the arrival of any new node, on the inductive subgraph over $\mathcal{V}_{\text{tr}} \cup \mathcal{V}_{\text{cal}}$. Prior to the evaluation of any node, we compute the calibration quantile $q$ from calibration scores. For any test node, the conformity score of all its classes is compared to $q$. Clarkson (2023) shows that this approach fails to provide valid prediction sets (the guarantee is broken due to the shift caused by the arrival of new nodes).

**Computational complexity**. Conformal prediction has two main computational routines in addition to model's inference: computing scores and quantiles. For the calibration set we only need the scores of the true class, but for test nodes, we should compute scores for all classes. Depending on the score function the complexity can be different. For standard CP we compute calibration scores once, but for NodeEx CP for each evaluation (timestep) we need to recompute the predictions and the conformity scores. Hence, Node(Edge)Ex CP has an overhead of $O(n \times t_s \times t)$ for $t$ timesteps, $n$ calibration nodes, and $t_s$ the time for computing a conformity score for one class and one node. At each step we should also evaluate the model once if the model's prediction changes across time. Computing the quantile threshold takes $\mathcal{O}(n)$ steps. For standard CP we have to compute this value once, but in NodeEx CP this values is updated upon each evaluation. The total wall-clock overhead is less the a second. Additionally all mentioned complexities are for serial computation and some of CP procedures can run significantly faster via parallel computation.

**Limitations of NodeEx and EdgeEx CP**. Our study mainly focuses on approaches that can provide a valid CP for changing subgraphs under certain assumptions of node-or edge-exchangeability. There are three main limitations for using NodeEx (or EdgeEx) CP (i) The guarantee provided by NodeEx (and EdgeEx) CP is marginal. However, it is shown that exact conditional coverage $\text{Prob}\left[y \in \mathcal{C}(\boldsymbol{x}) \mid \boldsymbol{x}\right]$ is not achievable. Still, there are methods that tend to get better approximations of conditional coverage. (ii) In real-life graph sequences, it is hard to determine whether the graph sequence is node- or edge-exchangeable or neither. Here a follow-up works is to use the beyond exchangeability approach (Barber et al., 2023) to achieve valid coverage guarantees without the need for node- or edge-exchangeability assumption. (iii) Our result maintains validity as long as the selection evaluation timestep for each node is done without any knowledge of the prediction set. An adversarial selection can cause a significant deviation from the guaranteed coverage. For instance, an adversary can observe prediction sets for a particular node during various timesteps and pick the timestep of the smallest prediction set similar to the two false examples in § D.1. This results can lead to a significant miscoverage rate compared to the guarantee.

## B    ADDITION TO THE THEORY

**Matric $C$.** For timestep $t$ and index $i$, $C[i, t]$ indicates that whether node $v_i$ if evaluated at timestep $t$ is going to be covered or not. Hence this value can be either 1, indicating CP covers node $v_i$ at timestep $t$, 0 showing that CP does not cover this node, or $N/A$ showing that this node has not appeared at this timestep. This matrix is constructed using ground-truth labels and is shown only for the purpose of sanity check and evaluation. The matrix $C$ is a very dense matrix since $1 - \alpha$ entities at each column should be 1. That is why in Fig. 2 we showed $1 - C$ where each column is expected to have $\alpha$-percentage of ones. A valid CP approach should result in a similar percentage of ones in each column. For an invalid CP, if the trend is gradually moving toward over-coverage, the matrix $1 - C$ becomes more sparse in later columns.

**Node- and edge- exchangeability**. A sequence $\mathcal{Z} = (z_1, \ldots, z_n)$ is called exchangeable if for any permutation $\mu$, the joint probability of the permuted sequence remains the same, i.e. $\text{Prob}[(z_1, \ldots, z_n)] = \text{Prob}[(\mu(z_1), \ldots, \mu(z_n))]$. Before considering inductive graph sequences, we first define node- and edge-exchangeability on a fixed given graph $\mathcal{G}$.

**Definition 1** (Node-exchangeability). *Let $\mathcal{G}(\mathcal{V}, \mathcal{E}, X, y)$ be a graph where $i$-th row of $X$ is the feature vector for node $v_i$, and similarly, $i$-th element of $y$ is the label of that node. Let $\mu_V : \{1, \ldots, n\} \times \{1, \ldots, n\}$ with $n = |\mathcal{V}|$ be a permutation. We define $\mu_V(\mathcal{G})$ to be a relabeling of vertices $\mathcal{V}$ with $\mu_V(\mathcal{E}) = \{(\mu_V(v_i), \mu_V(v_j)) : \forall (v_i, v_j) \in \mathcal{E}\}$, $\mu_V(X)[i, \cdot] = X[\mu_V(i), \cdot]$, and $\mu_V(y)[i] = y[\mu_V(i)]$. The permuted graph is $\mu_V(\mathcal{G})(\mu_V(\mathcal{V}), \mu_V(\mathcal{G}), \mu_V(X), \mu_V(y))$. $\mathcal{G}$ is called node exchangeable if $\text{Prob}[G = \mathcal{G}] = \text{Prob}[G = \mu_V(\mathcal{G})]$, where $G$ is sampled from a generative graph distribution.*

**Definition 2** (Edge-exchangeability). *Let $\mathcal{G}(\mathcal{V}, \mathcal{E}, X, y)$ where $\mathcal{E} = \{e_1, \ldots e_m\}$ is the set of edges. Each edge $e_k$ is defined as a pair $e_k = ((v_i, x_i, y_i), (v_j, x_j, y_j))$. For each node $v_i \in \mathcal{V}$ let $X$ be the feature matrix with rows $x_i$, and $y$ be the vector of labels with entries $y_i$. Let $\mu : \{1, \ldots, m\} \times \{1, \ldots, m\}$ with $m = |\mathcal{E}|$ be a permutation. We define $\mu(\mathcal{G})$ to be a relabeling of edges $\mathcal{E}$ with $\mu_E(\mathcal{E}) = \{e_{\mu_E(i)} : \forall e_i \in \mathcal{E}\}$. The permuted graph is $\mu_E(\mathcal{G})(\mathcal{V}, \mu_E(\mathcal{E}), X, y)$. $\mathcal{G}$ is called edge exchangeable if $\text{Prob}[G = \mathcal{G}] = \text{Prob}[G = \mu_E(\mathcal{G})]$, where $G$ is sampled from a generative graph distribution.*

By an inductive sequence, we refer to a progressive sequence of graphs meaning that for each timestep $t$, the graph $\mathcal{G}_{t-1}$ is a subgraph of $\mathcal{G}_t$. Node- and edge-inductive sequences are defined as follows:

**Definition 3** (Node-inductive sequence). *A node-inductive sequence $\mathcal{G}_0, \mathcal{G}_1, \ldots$ is a sequence starting from an empty graph $\mathcal{G}_0 = (\varnothing, \varnothing)$. For each timestep $t$, the graph $\mathcal{G}_t$ is defined by adding a vertex $v_t$ with all its connections to the vertices in the previous timestep. The vertex set is then $\mathcal{V}_t = \mathcal{V}_{t-1} \cup \{v_t\}$ and the edge set is the union of all previous edges $\mathcal{E}_{t-1}$ and any edge between $v_t$ and $\mathcal{V}_{t-1}$; $\mathcal{E}_t = \mathcal{E}_{t-1} \cup (\cap_{i=1}^{\infty} \{e = (v_t, v_i) : e \in \mathcal{E}_i, v_i \in \mathcal{V}_{i-1}\})$.*

Similarly an edge-inductive sequence is defined as follows:

**Definition 4** (Edge-inductive sequence). *An edge inductive sequence $\mathcal{G}_0, \mathcal{G}_1, \ldots$ starts from an empty graph $\mathcal{G}_0 = (\varnothing, \varnothing)$. For each timestep $t$, the graph $\mathcal{G}_t$ is defined by adding an edge $e_t = (v_i, v_j)$ to the graph at the previous timestep. Hence $\mathcal{E}_t = \mathcal{E}_{t-1} \cup \{e_t\}$, and $\mathcal{V}_t = \mathcal{V}_{t-1} \cup \{v_i, v_j\}$.*

A node-inductive sequence is called node-exchangeable if for all graphs $\mathcal{G}_t$ in the sequence, $\mathcal{G}_t$ is node-exchangeable. Similarly, if all graphs in the sequence are edge-exchangeable, the sequence is also edge-exchangeable.

Note, an inductive sequence could be equivalentlty defined with respect to a "final" graph $\mathcal{G}_n$, where $n$ can be infinity, by starting from an empty graph and adding a random node/edge at each timestep. If the final graph is node-exchangeable any node-inductive subgraph of it $\mathcal{G}' \subseteq \mathcal{G}_t$, $\mathcal{G}'$ is also node exchangeable. A similar argument holds for edge-inductive subgraphs of a "final" edge-exchangeable graph.

**Permutation invariance and equivariance**. A permutation invariant function $f$ returns the same output for any permutation applied on its inputs, $f(z_1, \ldots, z_n) = f(z_{\mu(1)}, \ldots z_{\mu(n)})$ for any permutation $\mu$. A function $f$ is permutation-equivariant if permuting the input results in the same permutation on the output, i.e. for any $f(x_1, \ldots, x_n) = (y_1, \ldots, y_n)$ we have $f(x_{\mu(1)}, \ldots x_{\mu(n)}) =$

$(y_{\mu(1)}, \ldots y_{\mu(n)})$ for any $\mu$. A permutation-equivariant GNN assigns the same predictions (and thus the same scores) to each node even when nodes/edges are relabeled.

*Proof for Proposition 1.* First assume $\mathcal{V}_{\text{eval}}^{(t)} = \mathcal{V}_t$ meaning that any node appeared at timestep $t$ is either in the calibration or the evaluation set. Due to node-exchangeability, $\mathcal{V}_t$ and $\mathcal{V}_{\text{cal}}$ are exchangeable. Hence, we adapt the proof of Theorem 2. Now consider the general case where $\mathcal{V}' := \mathcal{V}_t \setminus \mathcal{V}_{\text{cal}} \neq \emptyset$. Any subset $\mathcal{A}'$ of an exchangeable set $\mathcal{A}$ is still exchangeable. For any permutation of $\mathcal{A}'$ there are $k$ permutations in $\mathcal{A}$ all having the same ordering over elements of $\mathcal{A}'$. Since $k$ is a constant for all permutations and function of $|\mathcal{A}|$ and $|\mathcal{A}'|$, any permutation in $\mathcal{A}'$ has the same probability. Which implies the exchangeability of its elements. This implies that with non-empty set $\mathcal{V}'$, $\mathcal{V}_{\text{cal}}$ and $\mathcal{V}_t \setminus \mathcal{V}'$ are still exchangeable. As a result, again we adapt Theorem 2 with the effect of $\mathcal{V}'$ being symmetric to the calibration and evaluation sets. $\qquad\square$

*Proof for Theorem 3.* We break the proof to two parts.

**Part 1**. For a fixed node $v_i$ at timestep $t_1$ let $\alpha_i := \mathbb{E}[c_{i,t_1}]$. For any timestep $t_2$ with which $v_i$ is existing in, we have $\mathbb{E}[c_{i,t_2}] = \alpha_i$.

Note that since $v_i$ is fixed its expected coverage probability is not exactly $1 - \alpha$. As in definition,

$$\alpha_i = \text{Prob}\left[\text{Quant}\left(\alpha; \{s(v_j, y_j \mid \mathcal{G}_{t_1})\}_{v_j \in \mathcal{V}_{\text{cal}}}; 1\right) \leq s(v_i, y_i \mid \mathcal{G}_{t_1})\right]$$

Let $\mathcal{V}'_j = \mathcal{V}_{t_j} \setminus (\mathcal{V}_{\text{cal}} \cup \{v_i\})$. Node-exchangeability implies that $\mathcal{V}'_1$ and $\mathcal{V}'_2$ have same and also symmetric effect on all scores. Hence, with either of sets as context, the expected order of elements remains similar and $\mathbb{E}[c_{i,t_2}] = \alpha_i$. Moreover, for each node $v_i$ and all $t$, $c_{i,t} \sim \text{Bernoulli}(\alpha_i)$.

**Part 2**. The coverage of any sub-partitioning $\mathbb{I}_T$ can be written as an average over a set of elements in $\boldsymbol{C}_T$. In other words define $\iota(v_i) = j \Leftrightarrow (v_i \in \mathcal{V}_{\text{eval}}^j)$ we have

$$\text{Cov}(\mathbb{I}_T) = \frac{1}{|\mathbb{I}_T|} \sum_{i=1}^{|\mathbb{I}_T|} \boldsymbol{C}[i, \iota(v_i)] \stackrel{\text{Part 1}}{=} \sum_{i=1}^{|\mathbb{I}_T|} \boldsymbol{C}[i, T]$$

Following Proposition 1 we have

$$\text{Prob}\left[\text{Cov}(\mathbb{I}_T) \leq \beta\right] = \frac{1}{|\mathbb{I}_T|} \sum_{i=1}^{|\mathbb{I}_T|} \boldsymbol{C}[i, \iota(v_i)] = 1 - \Phi_{\text{HG}}(i_\alpha; |\mathbb{I}_T| + n, n, i_\alpha + \lceil |\mathbb{I}_T| \cdot \beta \rceil)$$

$\qquad\square$

*Proof for Theorem 4.* We divide the proof into two cases:

**Graph where all nodes have degree 1**. In this case $\mathcal{G}_t$ is a union of disjoint edges $(e_1, \ldots e_m)$ We divide $\mathcal{V}_t = \mathcal{V}_1 \cup \mathcal{V}_2$ including one and only one endpoint of each edge decided at random. The vertex set in this case has the same size as the edge set, and there is a one-to-one mapping between any permutation $\mu$ over the edge set to vertices in $\mathcal{V}_1$ or $\mathcal{V}_2$ since only one endpoint is present in each set. Hence $\text{Prob}[\mathcal{E}] = \text{Prob}[\mathcal{E}_\mu]$ implies $\text{Prob}[\mathcal{V}_i] = \text{Prob}[\mathcal{V}_{i,\mu}]_{i \in \{1,2\}}$. Similarly, selection $\mathcal{V}_{\text{cal}}$ decomposes to $\mathcal{V}_{\text{cal},1} \cup \mathcal{V}_{\text{cal},2}$ and again $\mathcal{V}_{\text{cal},i}$, and $\mathcal{V}_i \setminus \mathcal{V}_{\text{cal},i}$ are exchangeable due to node-exchangeability in $\mathcal{V}_i$. Any test node belongs to either of the subsequences which means is guaranteed via the CP applied to that subset.

The vertex set is divided into two subsets, and the intersection of the calibration set to each subset is exchangeable. Hence calibrating on each subset result in $1 - \alpha$ coverage for all the nodes in the subset. Both calibrations have the same expected quantile threshold $q$. This is because each calibration node has $1/2$ probability of being included in each of the sets.

**General graphs**. We follow the same approach. We divide the endpoints of edges into two multisets. For each edge we run this division and this decision is made for each vertex $\deg(v_i)$ times. In each partition, each node $v_i$ has the expected frequency $\deg(v_i)/2$. Via Lemma 1 we show that weighted CP is valid for each subsequence $\mathcal{V}_i$ and hence for $\mathcal{V}_t$. $\qquad\square$

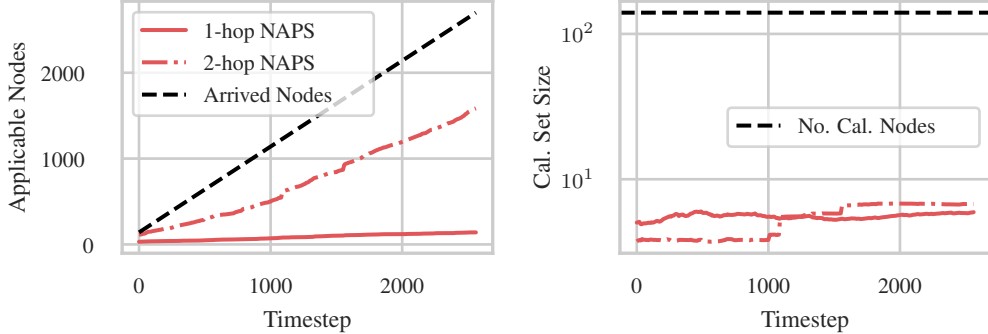

Figure 5: [Left] The proportion of nodes without a prediction set due to empty calibration on `CoraML` dataset. The plot shows the inapplicability of $k$-hop NAPS with $k \in \{1, 2\}$. [Right] The average size of each node's filtered calibration set. Note that this plot excludes non-applicable nodes.

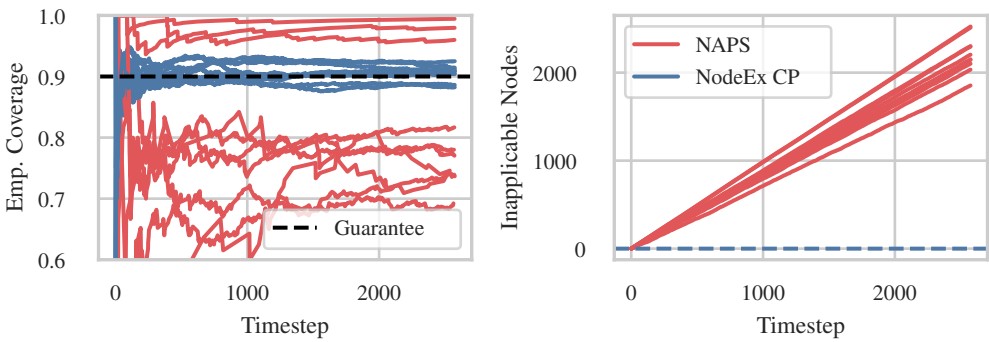

Figure 6: [Left] Comparison of NAPS and NodeEx CP in empirical coverage for `CoraML` dataset and `GCN` model across different permutations of nodes. [Right] The number of nodes without prediction set for different permutations on the same dataset/model.

## C  LIMITATIONS OF NAPS

For the inductive scenario, NAPS (Clarkson, 2023) adapts the beyond exchangeability approach (Barber et al., 2023) without any computed bounds on the coverage gap (the distance between the coverage of the given non-exchangeable sequence and $1-\alpha$). For each test node, calibration weights are assigned equal to $1$ in case they are in the immediate neighborhood of the test node and $0$ otherwise. In other words, this approach filters out the non-neighbor calibration nodes for each test node. This weight assignment is generalized to any $k$-hop neighbor while it is originally suggested to use $k = 1$ or $2$. In a sparse graph, or with a limited calibration set, the probability of having calibration points in the immediate neighborhood is significantly low which leaves many of the test nodes with empty calibration sets. Fig. 5 shows that for our experimental setup, until the end of the graph sequence, a notable proportion of the nodes are left without a prediction set. For the same reason, even for nodes in the neighborhood of the calibration set, the statistical efficiency is very low. Since this inapplicability is reported on a node-exchangeable sequence, it is independent of the prediction timestep – a node disconnected from the calibration set will remain disconnected as the graph grows.

Setting aside non-applicable nodes we compared NAPS with NodeEx CP in empirical coverage. Fig. 6 shows that NAPS is significantly far from the coverage guarantee compared to NodeEx CP. Note that this experiment is very biased in favor of NAPS as it excludes nodes without a prediction set while evaluating the same nodes for NodeEx CP.

## D ADDITIONAL EXPERIMENTS

### D.1 CP WITH SUBGRAPH SAMPLING

Although the theory in § 4 concerns inductive graph sequences, we can leverage its results to transductive node-classification via subgraph sampling. For each test node $v_{\text{test}}$, we sample $K$ inductive subgraphs all including $\{v_{\text{test}}\} \cup \mathcal{V}_{\text{cal}}$. Considering Theorem 3, in each subgraph, calibration scores and test scores are exchangeable. This leaves $K$ prediction sets all with a coverage guarantee of $1 - \alpha$. We consider each prediction set as a vote for all its elements. Each label $y'$ will be selected in the final prediction set with a Bernoulli experiment with parameter $p = \sum_{i=1}^{K} \mathbf{1}[y_j \in C_i(v_{\text{test}})]/K$. Again the resulting prediction set contains the true label with $1 - \alpha$ probability. It is important to note that Theorem 3 does not imply any result about the probability of a node being covered in all of $K$ prediction sets. Specifically, the probability of a node being covered in all prediction sets simultaneously is less than

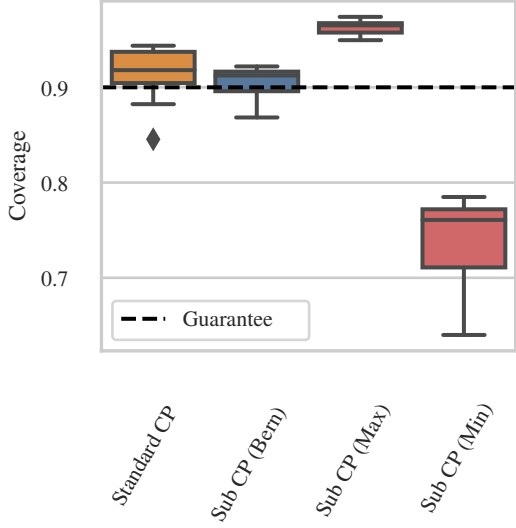

Figure 7: Comparison between standard CP, Bernoulli prediction sets from NodeEx CP on random subgraphs, the union, and intersection of CPs.

(or equal to) the probability of the same event in each single CP. As shown in Fig. 7 the subgraph Bernoulli prediction sets result in the same coverage as standard CP for transductive node-classification. Additionally, the union and intersection of CPs respectively result in over-coverage and under-coverage.

### D.2 SET SIZE AND SINGLETON HIT RATIO

In § 6 we focused on empirical coverage as the main evaluation criteria since our goal is to recover the conformal guarantee. When the guarantee holds, this metric is often reported as a sanity check. In that case, the average prediction set size (efficiency) is considered as a direct indicator of how efficient (and therefore useful) prediction sets are. Another important metric called *singleton hit ratio* is the frequency of singleton sets covering the true label. In Fig. 8 we show the coverage matrix alongside the prediction set size and singleton hits indicator for each node at each timestep. As shown in the figure, since the standard CP breaks the guarantee toward over-coverage, it results in larger prediction sets on average and hence, a lower number of singleton hits. The same result is also shown in Fig. 4 over different timesteps.

### D.3 DIFFERENT SCORE FUNCTIONS

The threshold prediction sets (TPS) approach (Sadinle et al., 2018) directly takes the softmax output as a conformity score $s(\boldsymbol{x}, y) = \pi(\boldsymbol{x})_y$, where $\pi(\boldsymbol{x})_y$ is the predicted probability for class $y$. Although TPS produces small prediction sets, its coverage is biased toward easier examples leaving the hard examples under-covered (Angelopoulos & Bates, 2021b). Romano et al. (2020) define *adaptive* prediction sets (APS) with a score function defined as $s(\boldsymbol{x}, y) := - (\rho(\boldsymbol{x}, y) + u \cdot \pi(\boldsymbol{x})_y)$. Here $\rho(\boldsymbol{x}, y) := \sum_{c=1}^{K} \pi(\boldsymbol{x})_c \mathbf{1}[\pi(\boldsymbol{x})_c > \pi(\boldsymbol{x})_y]$ is the sum of all classes predicted as more likely than $y$, and $u \in [0, 1]$ is a uniform random value that breaks the ties between different scores to allow exact $1 - \alpha$ coverage (Stutz et al., 2022). APS returns the smallest prediction sets satisfying conditional coverage if the model returns the ground truth conditional probability $p(y \mid \boldsymbol{x})$, otherwise Barber et al. (2019b) show that achieving conditional coverage is impossible without strong unrealistic assumptions. Here we use APS as baseline scoring function, but our method is orthogonal to this choice and works with any score.

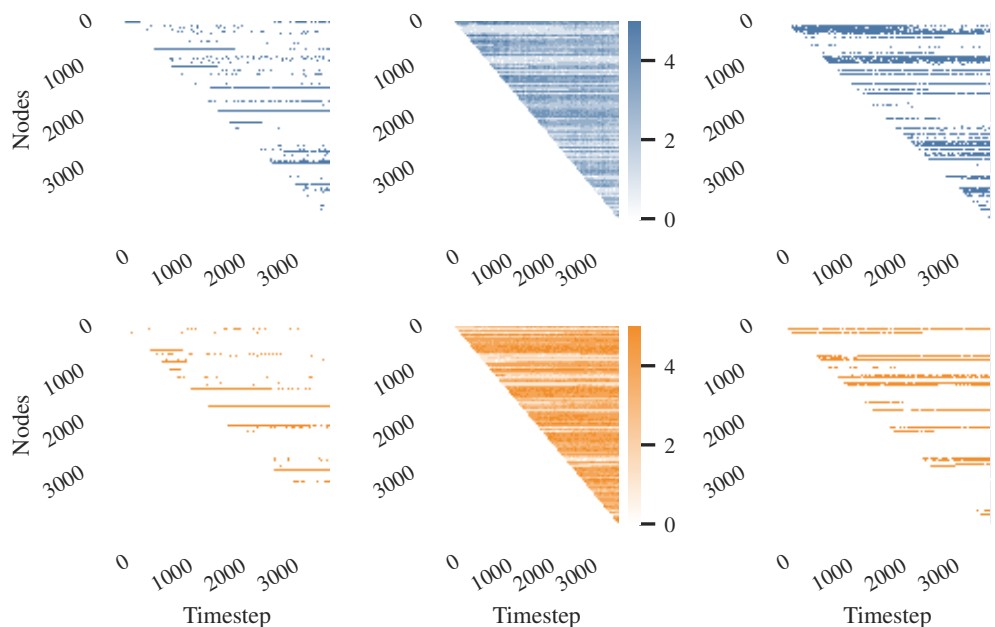

Figure 8: [Left column] The coverage matrix $1 - C$ (colored points show miscoverage). [Middle column] The prediction set size for each point at each timestep. [Right column The matrix of singleton hits. A cell is colored if the node at the timestep is predicted with a singleton set covering the true label. [Upper row] The NodeEx CP approach. [Lower row] The standard CP. The result is shown for `CiteSeer` dataset and `GCN` model.

Although we used APS as the base score function for evaluation, our results are general to any other scoring function. Here we evaluate our approach with two other score functions called threshold prediction sets (TPS) (Sadinle et al., 2018), and diffused adaptive prediction sets DAPS Zargarbashi et al. (2023). TPS (Fig. 9 - left) simply applies the softmax function on model's results (logits) and uses it as the conformity score. Although it produces small prediction sets its coverage is biased toward easier examples (Angelopoulos & Bates, 2021b). DAPS (Fig. 9 - right) works as a structure-aware extension scoring over APS. It diffuses conformity scores over the network structure

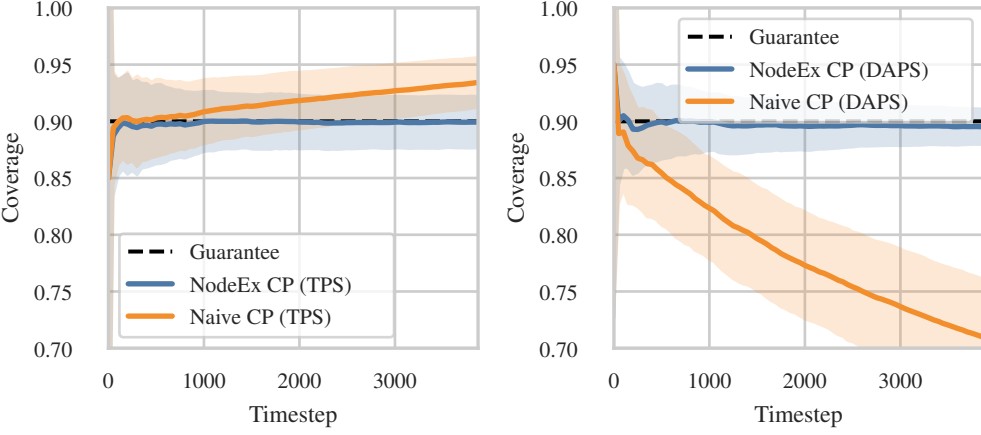

Figure 9: [Left] Comparison of naive CP and NodeEx CP with TPS score function and [Right] DAPS score function. Results are for `CiteSeer` dataset and `GCN` model.

to leverage the uncertainty information in the neighborhood and produce a more efficient prediction set. Since DAPS also incorporates the structure in the scores' space (in addition to the implicit effect via message passing) it is even more influenced by the changes in the graph structure while using standard CP. However, as shown in the Fig. 9, we can recover the coverage guarantee via NodeEx CP regardless of the utilized score function.

## D.4 DIFFERENT MODELS AND INITIAL SPLITS

The coverage guarantee in CP holds regardless of the model structure. CP uses the model as a black-box and just performs the quantile calibration over the output of the model which is the input conformity score function. However, better model structure, training procedure, etc are reflected in other metrics like set size and singleton hits. As shown in Fig. 10, NodeEx CP results in similar coverage values for all the models while standard CP results in different coverage for each model (note the significant distance between structure-aware models and MLP). As MLP does not take the adjacency structure into account, its empirical coverage is still guaranteed even with the standard CP. Note that different models result in different efficiencies in the prediction sets. Fig. 11 shows the same experiment conducted on edge-exchangeable sampling. Again similar results are observed for edge-exchangeable sequences.

As pointed out in (Shchur et al., 2018), GNNs are sensitive to the initial train/validation sampling. This however does not impact the coverage of NodeEx (and EdgeEx) CP as it is guaranteed agnostic to the model and the initial split. However similar to different model architectures, different initial splits also affect the model's accuracy which is reflected in the efficiency of the prediction set.

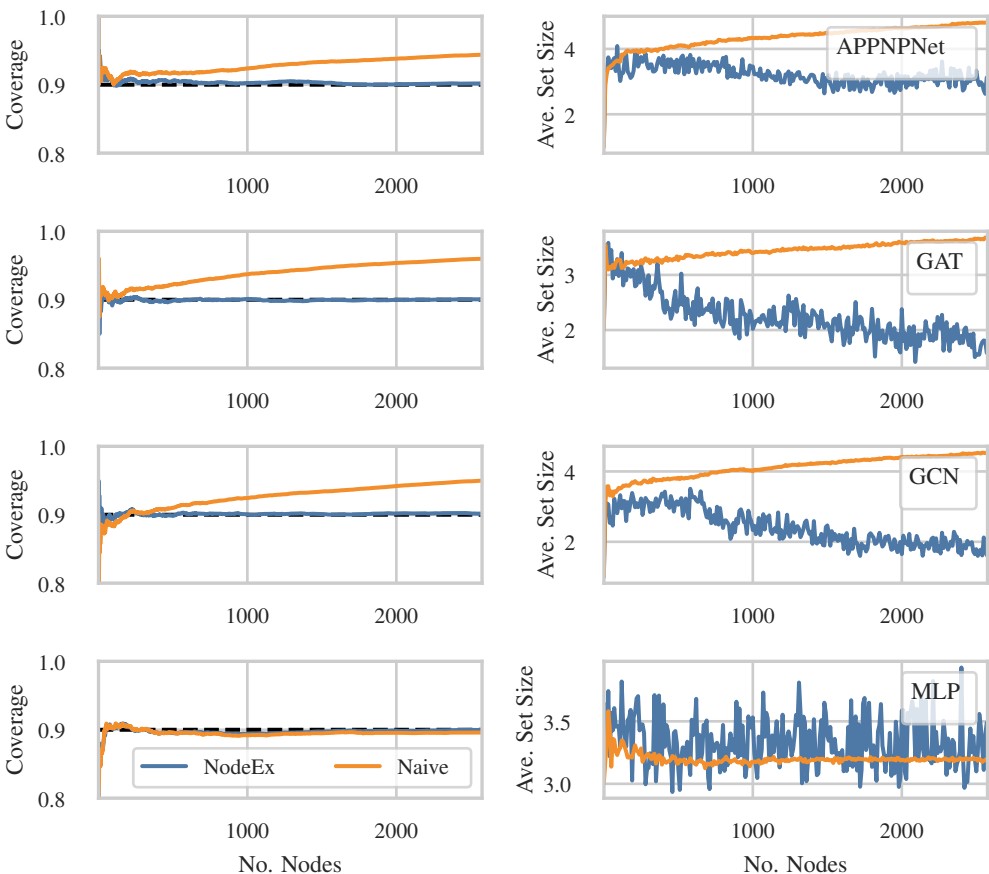

Figure 10: [Left] The empirical coverage for different models. [Right] The average set size. The results are shown for the `CoraML` dataset with node-exchangeable sampling.

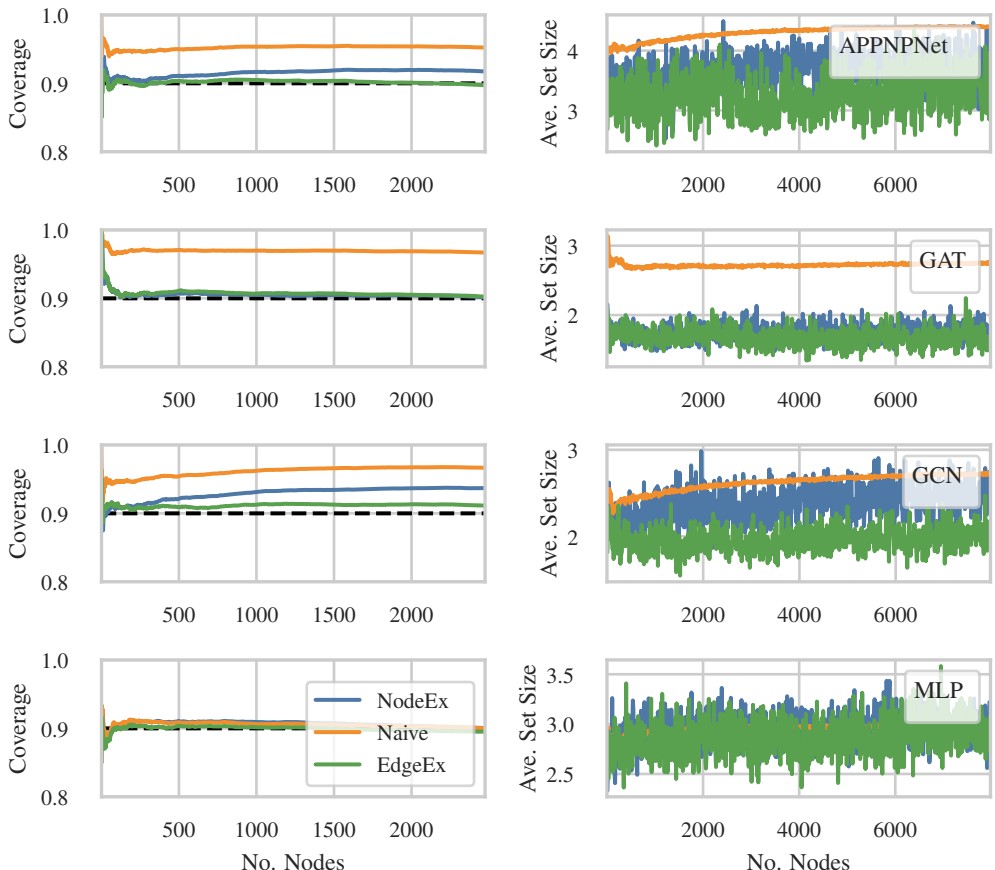

Figure 11: [Left] The empirical coverage for different models. [Right] the average set size. The results are shown for the `CoraML` dataset with edge-exchangeable sampling.

Fig. 12 verifies this fir the node-exchangeable sampling. We also evaluated our method compared to standard CP for edge-exchangeable sequences over different initial sampling. The result is in Fig. 13.

## D.5 OTHER EXPERIMENTS

**Experiment corresponding to Fig. 1**. We performed a node-exchangeable sampling on `CiteSeer` dataset. At each timestep, we ran the model (`GCN` trained on the train/val nodes) on the existing subgraph extracting conformal scores for all existing test nodes. The heatmap in Fig. 1 (left) shows the distribution of test scores (sorted) at each timestep. An oracle CP (with access to test nodes and their label) will choose the line labeled as "ground truth" as the conformal threshold since it is the exact $\alpha$ quantile of existing test scores. This is shown as the ideal reference to evaluate each CP approach with respect to it. We draw the threshold computed by standard CP and NodeEx CP. It is shown that the NodeEx CP picks thresholds close to the ideal line while the naive CP shifts from it due to the message passing with the new nodes.

We drew the distribution of conformity score for calibration nodes at some selected times (sampled with equal distance across from all timesteps) and compared it with calibration scores used by standard CP in Fig. 1(upper right). A distribution shift is clearly observable. We showed this shift in Fig. 1(upper right) by plotting the distibutions across various timesteps. The corresponding timestep for each distribution is marked in the left subplot with the same color. Here we computed the EMD (earth mover distance) between the scores of standard CP and conformity scores (of true labels) for calibration and test points. It is shown that the distribution shift is increasing by the number of new nodes introduced to the graph. This shift is almost similar in calibration and test. One source of

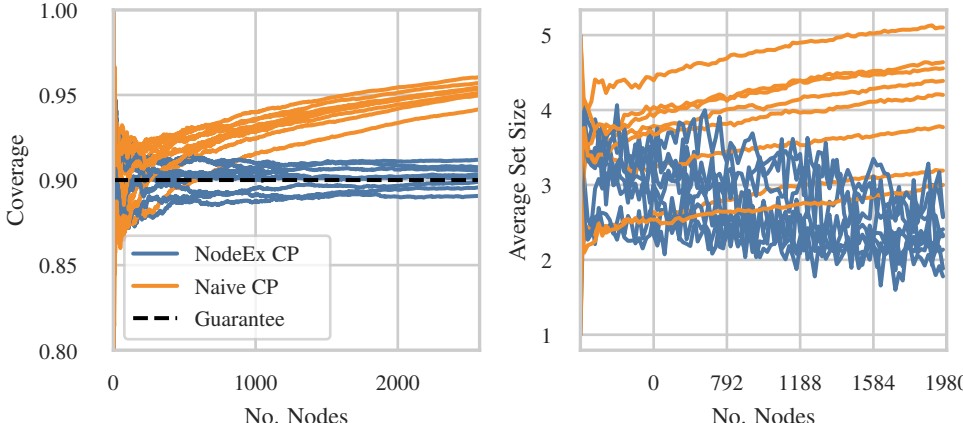

Figure 12: [Left] The empirical coverage of standard CP and NodeEx CP with different initial train/val splits. [Right] The average set size. The result is shown for `CoraML` dataset and `GCN` model for node-exchangeable samplings.

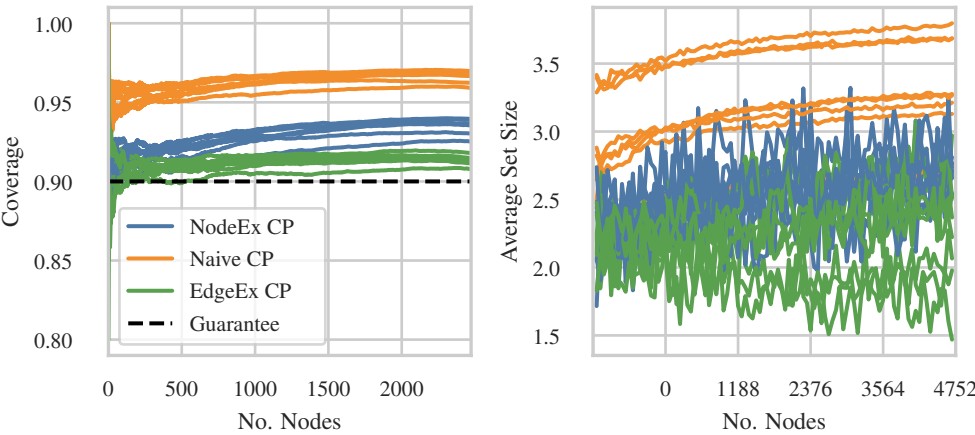

Figure 13: [Left] The empirical coverage of standard CP and EdgeEx CP with different initial train/val splits. [Right] The average set size. The result is shown for `CoraML` dataset and `GCN` model for edge-exchangeable samplings.

the noise in thresholds and EMD is the uniform random value in APS scoring function. EMD is smoothed over 10 steps.

**Large Datasets**. We ran NodeEx CP and compared it with the baseline for two large datasets: Flickr (Young et al., 2014), Reddit2 (Zeng et al., 2019). Our results for Flicker and Reddit2 datasets under node exchangeable sampling are shown in Fig. 14. Both NodeEx CP and naive CP show similar results. In the edge-exchangeable sampling Fig. 15 shows a significant deviation from the guarantee for NodeEx CP and naive CP, while EdgeEx CP maintains a valid coverage.

**Class-conditional coverage**. Conformal prediction comes with a marginal guarantee which means that it does not guarantee conditional to group or class. In Fig. 16 we compare this metric. In most of the classes, we observe the standard CP to be closer to the guaranteed line.

**Effect of calibration set size**. In non-graph data as mentioned in § 2 the coverage probability follows a Beta distribution. In graph data with a fixed number of test nodes (see Theorem 2) the distribution of coverage over test nodes follows a collection of hyper-geometric distributions. In both scenarios, there is a possible variance around predefined $1 - \alpha$ probability which is a function

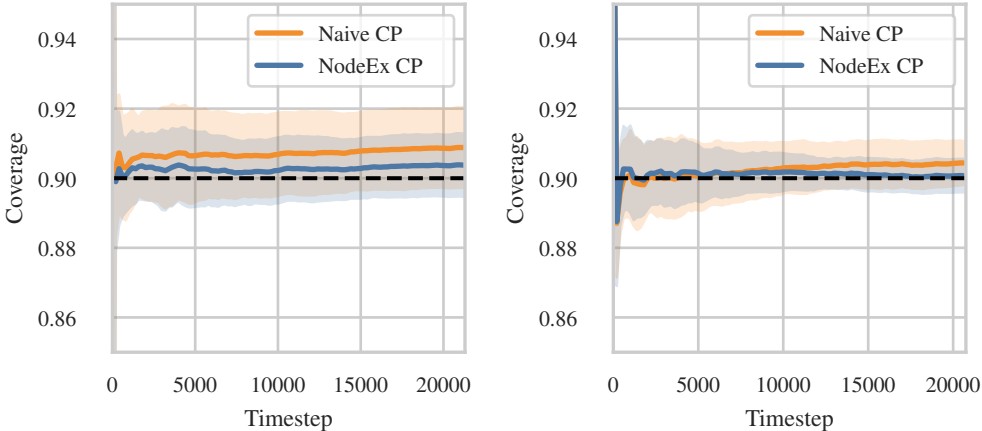

Figure 14: Coverage Results for [Left] `Flickr`, and [Right] `Reddit2` dataset under node-exchangeable sampling. The results are shown for `GCN model`.

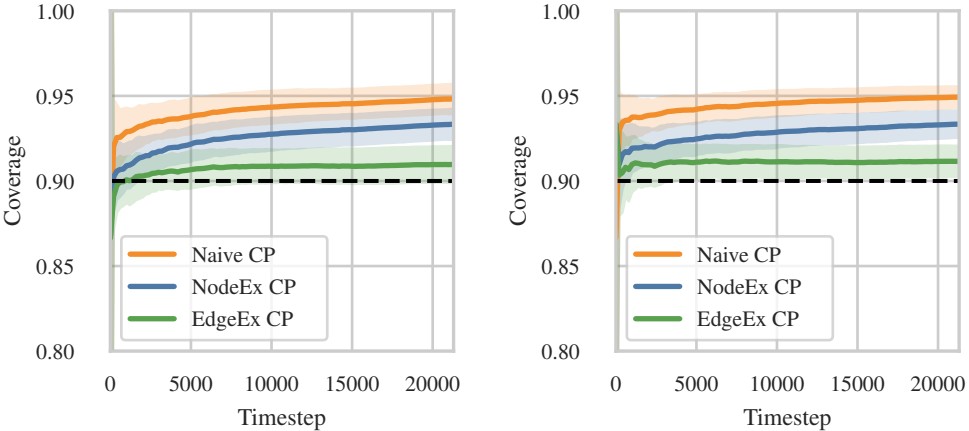

Figure 15: Coverage Results for [Left] `Flickr`, and [Right] `Reddit2` dataset under edge-exchangeable sampling. The results are shown for `GCN model`.

of calibration set size. As the number of calibration nodes increases, the distribution of coverage probability concentrates around $1 - \alpha$. Since in our experiments we followed a realistic setup (not allowing calibration set to be larger than training set) there is a variance observed around the guarantee line – each line converges to a value close to but not exactly equal to $1 - \alpha$. Fig. 17 shows that as we increase the calibration set size the result concentrates around the guarantee.

## E  SUPPLEMENTARY DETAILS OF EXPERIMENTS

**Datasets**. Table 2 provides specifications of datasets used in our experimental evaluations, including the number of nodes, edges, and homophily. The details of label sampling are provided in § 6. For each experiment, we ran each CP approach on 10 different sequences in node-exchangeable and 15 different sequences on edge-exchangeable setup. All transparent lines in plots show the result for one sequence. In those experiments, the solid line shows the average of sequences at each timestep.

**Models**. For all architectures, we built one hidden layer of 64 units and one output layer. We applied dropout on the hidden layer with probability $0.6$ for `GCN`, and `GAT`, $0.5$ for `APPNPNet`, and $0.8$ for

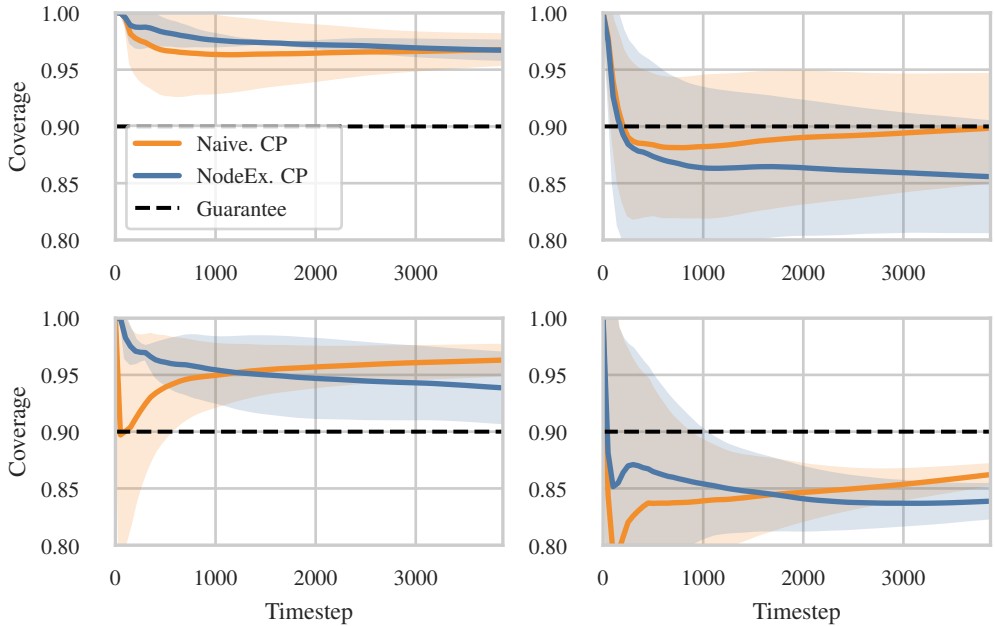

Figure 16: NodeEx CP and standard CP in class-conditional coverage. The result is for `CiteSeer` dataset and `GCN` model. Plots show classes 1 and 2 [Upper row left to right], 3 and 4 [Lower row].

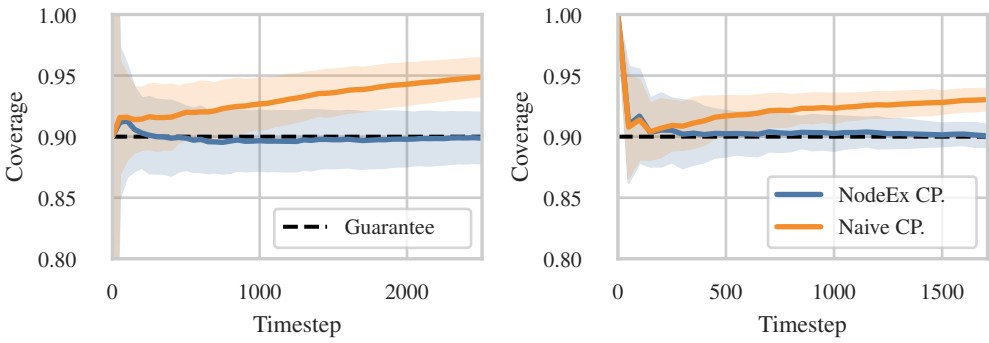

Figure 17: Effect of different calibration set sizes on the concentration of coverage probability. The results are on `Cora-ML` dataset and for 200 [Left] and 1000 [Right] calibration nodes.

`MLP`. For `GAT` we used 8 heads. We trained all models with categorical cross-entropy loss, and `Adam` optimizer with $L_2$ regularization.

**Adaptive and constant coverage**. Despite many studies on CP, Zargarbashi et al. (2023) chooses an adaptive value for $1 - \alpha$ which is conditional to the model accuracy. This is a suitable choice when the main comparison is based on set size and other efficiency-related metrics. The main supporting idea is that efficiency should be compared with a guarantee that is not trivially achievable by the normal model prediction. However, our main concern is to recover the guarantee meaning that the empirical coverage is the main comparison criteria. Hence we choose $0.9$ as the required coverage for all datasets and models regardless of the accuracy. Furthermore, NodeEx CP works for any user-specified coverage guarantee including model-conditional values.

Table 2: Statistics of the datasets.

| Dataset Name | Vertices | Attributes | Edges | Classes | Homophily |
|---|---|---|---|---|---|
| CoraML | 2995 | 2879 | 16316 | 7 | 78.85% |
| PubMed | 19717 | 500 | 88648 | 3 | 80.23% |
| CiteSeer | 4230 | 602 | 10674 | 6 | 94.94% |
| Coaut. CS | 18333 | 6805 | 163788 | 15 | 80.80% |
| Coauth. Physics | 34493 | 8415 | 495924 | 5 | 93.14% |
| Amz. Comp. | 13752 | 767 | 491722 | 10 | 77.72% |
| Amz. Photo | 7650 | 745 | 238162 | 8 | 82.72% |

## F    RELATED WORKS

**Standard conformal prediction**. CP is introduced by Vovk et al. (2005) and further developed in Lei & Wasserman (2014); Shafer & Vovk (2008); Barber (2020). Different variants of CP are yet defined ranging in the trade-off between statistical and computational efficiency. While full conformal prediction requires multiple training rounds for each single test point, Jackknife+ Barber et al. (2019a), and *split* conformal prediction sacrifice statistical efficiency, defining faster and hence more scalable CP algorithms. A comprehensive survey of CP can be found in Angelopoulos & Bates (2021a). In this study, we focus on split conformal prediction.

Further contributions in this area include generalization of the guarantee from including the true label to any risk function Angelopoulos et al. (2024), improving the efficiency (reducing the average set size) by simulating calibration during training Stutz et al. (2022), limiting the false positive rate Fisch et al. (2022), etc.

**CP without exchangeability**. Standard CP requires exchangeability for datapoints and assumes the model to treat datapoints symmetrically. The latter assumption ensures the exchangeability of datapoints even after observing the fitted model. Tibshirani et al. (2019) extend CP to cases where exchangeability breaks via different $p(X)$ for calibration set and the test data, given $p(Y \mid X)$ is still the same. In this case, CP is adapted via reweighing the calibration set using the likelihood ratio to compare the training and test covariate distributions. This requires the high-dimensional likelihood to be known or well-approximated. Barber et al. (2023) does neither rely on known likelihood between the original and shifted distribution nor the symmetry in the learning algorithm and proposes a coverage gap based on the total variation distance between calibration points and the test point. The main upperbound on the coverage requires a predefined fixed weight function. To adapt this bound to data-dependent weights, the $d_{TV}$-distance must be computed conditional to the assigned weights.

**CP for graphs**. Recently adapting CP to graphs got increasing attention. Wijegunawardana et al. (2020) adapted conformal prediction for node classification to achieve bounded error. Clarkson (2023) assumed exchangeability violated for node classification hence adapting weighted exchangeability without any lowerbound on the coverage, however, Zargarbashi et al. (2023) and Huang et al. (2023) proved the applicability of standard CP for transductive setting in GNNs. Moreover, Huang et al. (2023) proposed a secondary GNN trained to increase the efficiency of prediction sets, and Zargarbashi et al. (2023) stated that in homophily networks, diffusion of conformal score can increase the efficiency significantly.

For the inductive GNNs, the only work we are aware of is the neighborhood APS (NAPS) approach which is an adaptation of weighted CP Clarkson (2023), however, it is shown that NAPS can not be applied on a significant fraction of nodes if the network is sparse or the quantity of calibration nodes are limited Zargarbashi et al. (2023) (this number increases to over 70% in some benchmark datasets). As our approach is adaptable to sparse networks and works for any calibration set size, we do not compare our approach with NAPS.

