# OpenReview forum: "Conformal Inductive Graph Neural Networks"
_ICLR.cc/2024/Conference — ICLR 2024 poster_

### Official Review · Reviewer_aMS7 · 2023-10-23

**Soundness:** 2 fair
**Presentation:** 2 fair
**Contribution:** 3 good
**Rating:** 5
**Confidence:** 3

**Summary:**

The work develops conformal prediction for graph neural networks in the inductive setting (i.e., new nodes and edges are added in test time). Specifically, the authors address the implicit shift in calibration scores by recompute the calibration scores with respect to the new graph to restore coverage guarantees. Empirical comparisons against the standard CP (std. CP) on different GNNs and graph datasets are considered.

**Strengths:**

* Address the limitations of existing methods for inductive node-classification by recognizing that calibration and evaluation scores computed conditional to a specific subgraph are still exchangeable.
* Adapt the non-exchangeable CP method in the edge-exchangeable case, with weights inverse proportional to nodal degrees.
* Show promising results on different datasets and GNN models.

**Weaknesses:**

* Missing literature review: The work does not have a standalone literature review section.

* Writing:
   * Organization: the lengths between sections 4 and 5 are imbalanced. Specifically, section 4 contains the "method", theoretical analyses, and examples, where section 5 made reference to section 4 but the algorithm is delayed until the appendix. I think it makes more sense to first introduce the method, provide intuitions of why it works, and then give the theoretical analyses with additional remarks.
   * The Algorithm in Appendix A is not well-written. For example, it is unclear how $\boldsymbol{S}_t$ in line 5 is used later. The $\tau^t$ in line 11 is also different from that in lines 7 and 9. Furthermore, it is unclear where the algorithm starts because the line numbers for inputs are intertwined with those for algorithmic steps.

* Experiments:
    * As explained in the paper, it is expected that standard CP does not work (e.g., in terms of reaching the coverage guarantees). I expect to see additional baselines for this setting that can be compared.
    * Missing examples on larger graphs. For example, the Flickr and Reddit2 datasets in https://arxiv.org/pdf/2211.14555.pdf. These examples have considerably more nodes than the current settings.
    * Missing standard deviation of results. From Figure 3 and 4, it seems the standard deviations can be high (I assume each line with light color indicates results from a specific run).
    * Presentation needs to be improved. For example, the left column of Figure 3 and the Figure 4 have too many lines that affects readability. It is best to just show the average values with error bars. In addition, the authors considered 4 GNN models, but results for some more recent ones are only in the appendix.
    * Misleading Table 1: it says "average deviation from guarantee", but it does not mean a higher deviation necessarily means worse performance. This is because if deviation is positive, this indicates over-coverage, which is still acceptable (one just needs to lower the significance level). In fact, this seems to be the case from looking at Figure 3.

**Questions:**

* Method:
   * What is the complexity of the method with respect to graph size, number of samples, number of classes, etc. Specifically, if the graph $\mathcal{G}_t$ is large with complicated definition of score $s$, I expect line 5 of Algorithm 1 to create some computational burden in real-time use.
   * How does the method work if two or more nodes/edges are added at each time step?
* Experiments:
   * What exactly is the "standard CP" in the inductive context and how is it actually implemented? The idea of standard CP is mentioned in section 2, but that section is for general CP and transductive GNNs. I think the details should be discussed in the appendix with a reference in the main text, unless the baseline is taken directly from existing methods. Also, I think the abbreviation "std. CP" reads too much like standard deviation of CP, so I suggest changing the name.
   * Does the method always work well on any GNN models being considered? It is helpful and important to discuss the generality and/or limitation of the method for different GNN baselines
   * From looking at Figure 3 and 4, it seems standard CP coverages gradually increases as timestep increases. This leads to larger set sizes (as expected). Why is this the case? I think for a meaningful comparison, the empirical coverage should be set the same and then we can compare average set size.

---

> ### Author Response · Authors · 2023-11-20
>
> We thank the reviewer for reading our paper and comments. Here we address the concerns pointed out by the reviewer:
>
> **Weaknesses:**
>
> (1) Our approach was to discuss the related works in context throughout the paper. For completeness, we now added a dedicated section in the appendix (section F) and provide a forward reference in the main paper (end of section 2).
>
> (2) The suggestion to first introduce the method and then give the theoretical analyses is interesting and we agree that it can indeed improve readability. Since this is a major change that would disrupt the discussion with the reviewers (since sections will change) we defer this to after the rebuttal.
> Note, the algorithm which we added to the appendix for completeness is implicitly defined via the equations in section 4 since it essentially has only two steps: (1) computing (a potentially weighted) quantile line 3 or line 5, and (2) creating a set of scores above the quantile line 6. We also now updated it to clarify how $S_t$ is used.
>
> (3) We are not aware of any baselines that are applicable expect standard CP and NAPS. In Appendix C we compared our approach to NAPS. On Figure 6 (left) we show the empirical coverage and we see that our approach matches the nominal value while NAPS consistently under/overcovers. Note, here we outperform NAPS even though the experimental setup favours NAPS since we can only evaluate the coverage on the subset of applicable nodes (nodes that have at least one calibration node in their neighbourhood). On Figure 6 (right) we further show the number of non-applicable nodes for NAPS which grows with time. All nodes are applicable for our approach.
>
> As mentioned in section 2 and subsection 2.1, the coverage guarantee is probabilistic and can deviate from the specified value $1 - \alpha$. The concentration around $1 - \alpha$ (or the standard deviation) is a function of the calibration set size. The distribution (including the variance) is precisely characterized in Theorem 3. A realistic assumption for sparse graph is to have a calibration set not larger than the train set. Hence the small calibration set results in a relatively large variance. As we increase the calibration set size the variance quickly decreases (see figure 17).
>
> We added results on large graphs Flickr, and Reddit2 to the appendix (section D.5). Interestingly the results on node-exchangeable sampling is close for both standard and subgraph CP. However in the edge-exchangeable sampling the result is different and only weighted subgraph CP can provide a valid coverage guarantee.
>
> (4) As we explained in the paper, the coverage guarantee is bounded by upper- and lower-bounds. The coverage probability either follows a beta distribution, or a collection of hyper-geometric distributions which in both cases a valid CP results in a coverage close to $1 - \alpha$. An invalid guarantee can result in either over- or mis-coverage. There are two important notes: (i) an over-covering break of the guarantee results in larger set sizes and less singleton hits (see Figure 4) (ii) if the guarantee is invalid, one can not spot whether over- or mis-coverage happens before evaluating the result which makes it inapplicable. That is why in we reported the deviation from the guarantee showing that validity of the guarantee not the empirical coverage.

---

> > ### Author Response · Authors · 2023-11-20
> >
> > **Questions:**
> >
> >  - Method:
> >
> > (1) We added a discussion on the computational complexity in Section A (appendix). Mainly the added computational cost in subgraph CP is that the quantile threshold and the calibration scores should be recomputed upon each update in the graph.
> >
> > (2) The guarantees hold for any number of added vertices or edges. Essentially, we we would then only recompute the scores/quantile after $k$ updates.
> >
> >  – Experiments:
> >
> > (1) By the standard CP we refer to the same procedure as applied in [1]. In the standard CP, we compute the calibration scores and the threshold once the calibration nodes are present. While new nodes appear, we only compute the conformity scores for them and compare it with the threshold computed prior to the updates. We added an explanation of standard CP in the appendix (Section A).
> >
> > (2) Any GNN or model that is permutation-equivariant maintains exchangeability. Also there are other methods with focus on increasing the efficiency leveraging the network structure. As an example [2] diffuses the conformity scores, and [3] trains a GNN for the same purpose. Both methods are applicable on top of subgraph CP. At the end of section 6 we extended the discussion on limitations of our approach. We also examined the performance of other models in Section D.3.
> >
> > (3) Figure 3 reports empirical coverage which we cannot control or “set the same”. We set the nominal value $1-\alpha$ which is the same for CP and subgraph CP and then we can observe the empirical coverage over time.  Now, one might say that we could lower the nominal $1-\alpha$ for CP to counter the overcovering. However, in real-world cases we cannot compute empirical coverage because in practice we don’t have access to the labels for test nodes, and we don’t know if CP will overcover or undercover.
> >
> > **References**
> >
> > [1] Clarkson, Jase. "Distribution free prediction sets for node classification." International Conference on Machine Learning. PMLR, 2023.
> >
> > [2] Zargarbashi, Soroush H., Simone Antonelli, and Aleksandar Bojchevski. "Conformal Prediction Sets for Graph Neural Networks." (2023).
> >
> > [3] Huang, Kexin, et al. "Uncertainty quantification over graph with conformalized graph neural networks." arXiv preprint arXiv:2305.14535 (2023).

---

### Official Review · Reviewer_w8hy · 2023-10-28

**Soundness:** 3 good
**Presentation:** 3 good
**Contribution:** 3 good
**Rating:** 8
**Confidence:** 4

**Summary:**

This paper studies conformal prediction (CP) on graph data with graph neural networks in node-exchangeable and edge-exchangeable settings. The author(s) describe how certain exchangeability structure for nodes still persist in these settings, and how conformal prediction techniques can be adapted based on these observations. In particular, the proposed CP methods are valid conditional on subgraphs and when stopping at non-adversarial times, especially in cases when standard CP fails to provide the desired coverage. The methods and theory are clearly supported by empirical experiments.

Overall, I think this paper provides interesting discussion, which both extends the current understanding of CP on graph-structure data and forms a solid foundation for future efforts in this direction. Though the methods only adapt existing ones, the discussion on exchangeability structure is important. The presentation is overall clear, but potential improvements in mathematical rigor and presentation clarity would further strengthen this paper.

**Strengths:**

1. Significance of contribution.

Conformal prediction on graph data has just recently become a popular topic. This topic is important for deploying GNNs in critical domains. While existing discussion mainly focuses on transductive setting, or only provides approximate and not-actionable guarantees for other settings, this paper stands out by providing solid theory, method, and experiments on node-exchangeable and edge-exchangeable settings. By thoroughly exploring the exchangeability structure in these settings, existing CP methods can be applied to solve seemingly challenging problems in this field. Thus, I think this paper makes a significant contribution to this topic, and will inspire future research efforts of the community.

2. Originality.

The idea of this paper is original and provides novel insights.

3. Quality.

This is a thorough paper with solid theory, fruitful discussion, and concrete experiments.

4. Clarity.

The presentation is moderately clear. The positive points include (1) informative comparison to the literature, (2) useful remarks from place to place, and (3) succinct and clear statements of problem settings. I will mention points for improvement in the weakness part.

**Weaknesses:**

My complaints are not detrimental to this paper. I'm happy with the main ideas of this paper, so they are just comments that should be addressed to make the paper stronger.

Conformal prediction relies crucially rigorous mathematical statements about how data is generated / how test data appears in the set, how score functions are defined, how the score is trained and applied. These are all important factors for the validity of CP. While I believe the technical discussion in this paper is correct, sometimes I have to further check the statements in my own mind as some crucial mathematical terminologies are missing. A few possible points that deserve clarification include (1) formal definition of node and edge exchangeability, (2) clearly state the conditions on the score function and how the model is trained, (3) formal definition and examples of the "evaluation mask", etc. Please see my questions part. I'd be happy to increase my score if these points can be addressed.

**Questions:**

1. Mathematical definition of node exchangeability and edge exchangeability.

I suggest adding a formal definition for node and edge exchangeability, just to avoid any confusion. It will also be beneficial to mention some important implications of such exchangeability, e.g., at each time $t$, the existing nodes are exchangeable, so on and so forth.

2. Rigorous terminology in explaining Proposition 1.

Currently the explanation via $Z=AXW$ after Proposition 1 can be improved by using rigorous terminologies. For instance, in "For the guarantee to be valid the first two blocks of the result must be row-exchangeable.", it is unclear what "the first two blocks of the result" is, which is harmful for understanding. Also, related to question 1, "For Mat(1) row-exchangeability already holds from Theorem 2" can be further clarified by pointing out that nodes in cal and eval are exchangeable due to node exchangeability. I suggest the authors revise this part to improve clarity.

3. How is the evaluation set $\mathcal{V}_{eval}^{(t)}$ decided?

I am a bit confused how one decides the evaluation set at each time. Is there a pre-defined rule, for instance, do we evaluate all newly arrived nodes immediately at each time $t$, or do we pre-fix the set of indices that will be evaluated at each time $t$? I was asking because if $\mathcal{V}_{eval}^{(t)}$ is data-dependent, it may need more efforts in justifying the approach/theory.

4. Definition of $Cov(\mathbb{I}_T)$.

This is an important quantity in the theorem, and it shall be formally defined.

5. Requirement on score function in Prop 1 and Theorem 3.

It seems to me that under node exchangeability, one still has exchangeability between calibration and evaluation nodes. However, it is now crucial how the score function $s$ is trained. At least it should be permutation invariant to these nodes? But nothing is mentioned there. I suggest revising this proposition (and its proof, which is a bit handwavy). Similarly for Theorem 3.

6. Is there a technical result for the validity of weighted CP in edge-exchangeable setting?

The discussion for edge exchangeability (the end of Section 4) is interesting, but incomplete as there is no formal theorem on the validity of weighted CP. I would suggest adding a formal theorem, with clear description on how the edge appears (e.g., it is more clear to state that edges are uniformly sampled from a candidate edge set).


-------
small typos & wording questions:
1. Section 4, line 3: "callibration node" should be "calibration node"

2. Lemma 1, "defined on a subsets of $\mathcal{X}$"

3. I don't really understand the sentence "Specifically we can not use the prediction sets of a particular node multiple times to compare and pick a specific one (see § D.4 for a discussion)." in the second paragraph of Section 5.

**Details Of Ethics Concerns:**

none.

---

> ### Author Response · Authors · 2023-11-20
>
> We thank the reviewer for constructive comments and we are happy that the reviewer liked the paper. Here we address the comments and questions:
>
> (1) We added a formal description of node-inductive and edge-inductive sequences and node-exchageability and edge-exchageability in section B of the appendix and modified sections 4.1 and 4.2. All changes are shown with blue color in the updated paper.
>
> (2) Thank you for the valuable suggestions. We changed some parts of section 4.1 which hopefully make the example clearer.
>
> (3) Which node is included at which timestep is a design choice. One can wait until the last timestep and evaluate all the nodes at once, or evaluate each node upon arrival, or $k$ steps after arrival, etc. This corresponds to looking at e.g. columns of the coverage matrix $\boldsymbol{C}$, or the different diagonals of matrix C. As long as the decision is made prior to observing the prediction sets (independent of the data) the coverage validity holds.
> Importantly, this implies that the guarantee does not hold for adversarially chosen evaluation sets. To illustrate this, we can pick the smallest prediction set for each node across all timesteps. This breaks the guarantee as shown in Figure 13 (Section D.4 in appendix).
>
> (4) Thank you for this suggestion. We now define $\mathrm{Cov}(\cdot)$ in section 2.1.
>
> (5) Indeed, we assume that the underlying model and thus the score function is permutation-equivariant. We updated our papers and the theorems to clarify this. We also discuss this assumption in more detail in appendix B.
>
> (6) In light of the reviewer’s comment, we revised section 4.2 and added a formal theorem and a proof (section B in the appendix). In the paper we assume that the nodes from the first $m$ edges from an edge-exchangeable sequence are chosen for the calibration set. Any other exchangable selection that does not depend on the edge order, e.g. uniform sampling, is valid.
>
> We also thank the reviewer for mentioning the typos. We have resolved them in the modified file.

---

### Official Review · Reviewer_daSk · 2023-10-31

**Soundness:** 3 good
**Presentation:** 2 fair
**Contribution:** 3 good
**Rating:** 6
**Confidence:** 2

**Summary:**

While for transductive node-classification, we are given a fixed graph, in the inductive scenario the graph is changing. This leads to a shift in the scores and the standard conformal prediction methods do no longer apply.  This paper therefore proposes a method for calculating conformal prediction sets within the framework of inductive graph classification. The authors demonstrate coverage guarantees and provide extensive empirical experiments to show that their method is effective.

**Strengths:**

This is an interesting conformal prediction problem.

The paper seems solid with a large amount of experiments.

The code of experiments is available in the supplementary material.

**Weaknesses:**

1\ I found the paper difficult to follow. For instance, the setting of transductive node-classification of Section 2.1 or the inductive one of Section 3 are not properly defined. Another example is Fig.1 which is, in my opinion, not very clear and not explained properly. The definition of a graph is never given etc. Globally, the paper lacks explanation.

2\ There is a misuse of several words/expressions to discuss known concepts, which ultimately confuses the paper. What is the added value of a phrase such as "calibration budget" to mean that we only use the first N nodes or "no longer congruent with test scores" to mean that it is no longer exchangeable?

3\ The paper never indicates when scores should be assumed to be continuous in theoretical results. For instance, in all the remarks below Theorem 1, we need this assumption.

4\ There is no "limitation" in the conclusion.

Minor:

typos "callibration", "hyper-geomteric"

**Questions:**

Is it possible to "boost" any given non-conformity score with graphical information as in "Uncertainty Quantification over Graph with Conformalized Graph Neural Networks"?

Can you give the mathematical definition of "node-exchangeability" and "edge-exchangeability"?

What are the main limitations of the method?

Can you explain Fig1 and Fig9 in more detail?

---

> ### Author Response · Authors · 2023-11-20
>
> We thank the reviewer for the time and constructive comments. In following we address the mentioned concerns:
>
> **Weaknesses**
>
> (1) Thank you for these valuable suggestions. We added the definition of the graph, and the assumptions on the trained model in section 2.1. We also added formal definitions of node-inductive and edge-inductive sequences, as well as node- and edge-exchangeability. We updated the caption of Fig. 1 to improve clarity and included a forward pointer to the appendix (D.5) where the experiment is explained in full detail.
>
> (2) We agree that the term "calibration budget" does not bring additional value so we removed it from the paper. We also agree that the term "no longer congruent" is not necessary and have replaced it with "no longer exchangeable".
>
> (3) You are right, thank you. We modified Theorem 1 and the text to reflect this.
>
> (4) See the answer to question 3.
>
> **Questions:**
>
> (1) Yes. Both [1] and [2] propose methods to boost conformal prediction on graphs. [1] approaches this problem by a diffusion over the conformity scores, and [2] trains another GNN network from scratch. We theoretically show that any continuous score function while applied with subgraph (and weighted subgraph) CP can adapt to node- or edge-exchangeable inductive graph sequence. Any boosting method on top remains valid if applied in our framework.
>
> (2) Thank you for this suggestion. We modified the beginning of section 2.1, 3, 4.1, and 4.2 and included a more formal definition of node- and edge-exchangeability. All changes are shown with blue color in the updated paper. Additionally we dedicated a discussion in appendix B.
>
> (3) We identified three main limitations. First, as with most other CP methods, the guarantee is marginal. Second, real-world graph may not satisfy node- or edge-exchagability. This can be partially mitigated by the beyond exchangeability framework. Finally, the guarantee does not hold for adversarially chosen evaluation sets $\mathcal{V}_\mathrm{eval}$. In other words, the choice of which nodes and at which timesteps are included in the evaluation set must be done prior to observing the prediction set. In the initial submission we only mentioned the first limitation. We now added a discussion of all three (last paragraph of section 6).
>
> (4) We updated the caption of Fig. 1 to improve clarity and included a forward pointer to the appendix (D.5) where the experiment is explained in full detail.
> Figure 9 compares Subgraph CP with Standard CP for different GNN models. CP is model-agnostic, and although the model architecture (as long as it is permutation equivariant) does not affect the validity, it does affect the set size. On the left we show empirical coverage which matches the nominal value for our approach for all models. The MLP baseline does not use the graph structure and is unaffected by the distribution shift. On the right we see that some GNNs return smaller sets on average.
>
> **References**
>
> [1] Zargarbashi, Soroush H., Simone Antonelli, and Aleksandar Bojchevski. "Conformal Prediction Sets for Graph Neural Networks." (2023).
>
> [2] Huang, Kexin, et al. "Uncertainty quantification over graph with conformalized graph neural networks." arXiv preprint arXiv:2305.14535 (2023).

---

> > ### Comment · Reviewer_daSk · 2023-11-22
> >
> > I would like to thank the authors for their detailed response to my concerns.

---

### Official Review · Reviewer_2Ri3 · 2023-11-08

**Soundness:** 2 fair
**Presentation:** 2 fair
**Contribution:** 3 good
**Rating:** 6
**Confidence:** 3

**Summary:**

This paper explores the application of conformal prediction to graph neural networks, particularly addressing the challenge of inductive settings where the entire graph is not visible. In such scenarios, classic conformal prediction cannot be directly applied due to the loss of exchangeability when new nodes are introduced, which shifts the distribution of calibration scores.

**Strengths:**

The key insight of this paper is to demonstrate that the impact of new data points on the calibration and evaluation sets is symmetric, ensuring that the shift embedding still maintains its guarantee, conditional to the graph at a specific time step.

**Weaknesses:**

(1) Notations:
The presentation of this paper leaves room for improvement, as some notations are either omitted or unclear, making it misleading for readers. For instance, in Equation (3), the graph is not explicitly noted, and on page 5, in the linear message passing example, the meanings of A, X, and W are not immediately clear. Although these aspects become clearer upon reading the entire paper, the initial lack of clarity can impede readers from following the logic in their first encounter.

(2) Baselines:
In the experimental section, the paper exclusively utilizes standard CP as the baseline. While the authors explain the rationale for not including NAPS as a baseline, it would be beneficial to at least include NAPS results or another inductive-case baseline in the experimental evaluation for a more comprehensive assessment.

(3) Efficiency:
For CP methods, it's essential to consider both validity and efficiency. While the experimental results focus on "coverage," which assesses validity, it's equally crucial to assess efficiency. A predictive set that is overly large may not be practically useful. Therefore, it is recommended to include at least one experiment that demonstrates the efficiency of the subgraph CP method.

 Taking into account the feedback from other reviewers and the author's response, I have decided to raise my score.

**Questions:**

What is the motivation behind presenting Figure 2? Please consider adding an introductory explanation for this visualization figure.

---

> ### Author Response · Authors · 2023-11-20
>
> We thank the reviewer for the time and constructive comments on our paper. Here we address questions and concerns:
>
> **Weaknesses:**
>
> (1) We added the explicit definition of the graph in section 2.1. We also added the assumptions for the trained model (e.g. GNN), and the notation of graph in equation (3) follows the same definition. The graph $\mathcal{G}_t$ here shows the timestep in which the scores are computed (it refers to the present nodes and edges at timestep $t$).
> For the definition of simple message passing we added the definition of each matrix. Here $\boldsymbol{A}$ refers to the adjacency matrix , and $\boldsymbol{X}$ is the feature matrix (defined in section 2.1) and $\boldsymbol{W}$ refers to the weight matrix  which is defined immediately after.  We thank the reviewer for mentioning that and in case the notation is still not clear we ask the reviewer to kindly point it out.
>
> (2) In Appendix C we compared our approach to NAPS. On Figure 6 (left) we show the empirical coverage and we see that our approach matches the nominal value while NAPS consistently under/overcovers. Note, here we outperform NAPS even though the experimental setup favours NAPS since we can only evaluate the coverage on the subset of applicable nodes (nodes that have at least one calibration node in their neighbourhood). On Figure 6 (right) we further show the number of non-applicable nodes for NAPS which grows with time. All nodes are applicable for our approach.
>
> (3) We draw the reviewer’s attention to Figures 4, 7 (middle, right), 9 (right), 10 (right), 11 (right), 12 (right). We report two metrics regarding efficiency: average set size, and singleton hit ratio which captures what proportion of test nodes are predicted with a singleton set the includes the true label. On all figures and all metrics our approach is superior and has lower average set size and higher singleton hit ratio.
>
> **Question:** The idea of figure 2 is to show which nodes are (mis)-covered at which timestep. We added description of matrix C and Figure 2, in Section B (appendix) with a reference to it in the caption of the figure. We also draw the review’s attention to Figure 7 (in the appendix), where we compare both methods standard CP and subgraph CP. In the same figure we also compare the matrix of set sizes and singleton hits.

---

### Meta-Review · Area_Chair_K1bX · 2023-12-07

**Metareview:**

This paper addresses the conformal prediction for graph neural networks in the inductive setting. All of reviewers agree that the idea seems to be sound and interesting. Most of reviewers criticized some weaknesses in both writing and experiments. However, the authors well responded to these concerns, improving the paper during the rebuttal period. This led that two of reviewers increased their scores. This work may be interesting to audience working on conformal prediction or graph neural networks.

**Justification For Why Not Higher Score:**

I do not mind if it is bumped up to  "accept with spotlight".

**Justification For Why Not Lower Score:**

The limitations of existing methods are well recognized and it presents a nice idea to solve it.

---

### Decision · Program_Chairs · 2024-01-16

Accept (poster)